# Incremental Spatial and Spectral Learning of Neural Operators for Solving Large-Scale PDEs

**Robert Joseph George**                                                          *rgeorge@caltech.edu*
*Department of Computing and Mathematical Sciences*
*California Institute of Technology*

**Jiawei Zhao**                                                                   *jiawei@caltech.edu*
*Department of Computing and Mathematical Sciences*
*California Institute of Technology*

**Jean Kossaifi**                                                                 *jkossaifi@nvidia.com*
*NVIDIA AI*

**Zongyi Li**                                                                     *zongyili@caltech.edu*
*Department of Computing and Mathematical Sciences*
*California Institute of Technology*

**Animashree Anandkumar**                                                         *anima@caltech.edu*
*Department of Computing and Mathematical Sciences*
*California Institute of Technology*

**Reviewed on OpenReview:** *https://openreview.net/forum?id=xI6cPQObp0*

## Abstract

Fourier Neural Operators (FNO) offer a principled approach to solving challenging partial differential equations (PDE) such as turbulent flows. At the core of FNO is a spectral layer that leverages a discretization-convergent representation in the Fourier domain, and learns weights over a fixed set of frequencies. However, training FNO presents two significant challenges, particularly in large-scale, high-resolution applications: (i) Computing Fourier transform on high-resolution inputs is computationally intensive but necessary since fine-scale details are needed for solving many PDEs, such as fluid flows, (ii) selecting the relevant set of frequencies in the spectral layers is challenging, and too many modes can lead to overfitting, while too few can lead to underfitting. To address these issues, we introduce the *Incremental Fourier Neural Operator* (iFNO), which progressively increases both the number of frequency modes used by the model as well as the resolution of the training data. We empirically show that iFNO reduces total training time while maintaining or improving generalization performance across various datasets. Our method demonstrates a 38% lower testing error, using 20% fewer frequency modes compared to the existing FNO, while also achieving up to 46% faster training and a 2.8x reduction in model size.

## 1 Introduction

Recently, deep learning has shown promise in solving partial differential equations (PDEs) significantly faster than traditional numerical methods in many domains. Among them, *Fourier neural operator* (FNO), proposed by Li et al. (a), is a family of neural operators that achieves state-of-the-art performance in solving PDEs Azizzadenesheli et al. (2024), in applications modeling turbulent flows, weather forecasting Pathak et al. (2022), material plasticity Liu et al. (2022), and carbon dioxide storage Wen et al. (2022). FNO is specifically designed to learn mesh-independent solution operators for PDEs. Unlike traditional neural networks that learn point-wise mappings, FNO learns a functional mapping between infinite-dimensional

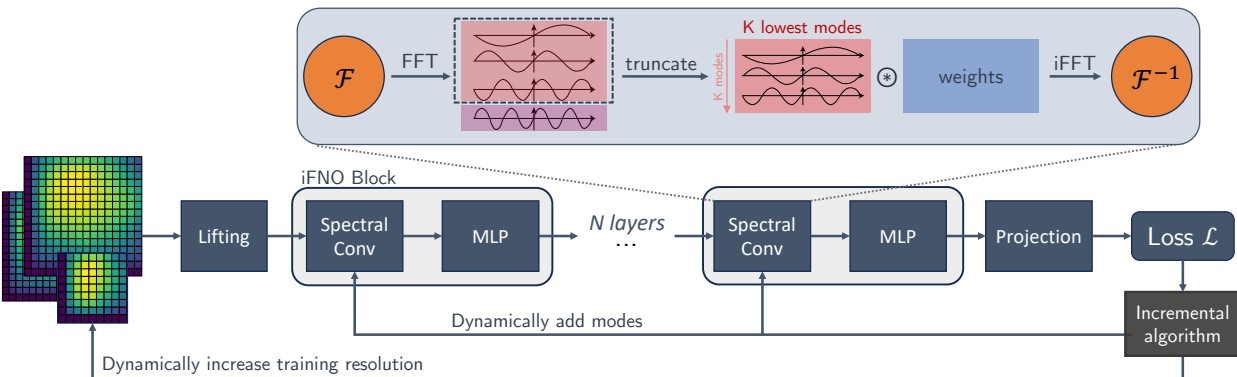

Figure 1: **Top: Fourier convolution operator in FNO.** After the Fourier transform $\mathcal{F}$, the layer first truncates the full set of frequencies to the $K$ lowest ones using a dynamically set truncation before applying a learnable linear transformation (*blue*) and finally mapping the frequencies back to the linear space using the inverse FFT $\mathcal{F}^{-1}$. The previous method (Li et al., a) picks a fixed $K$ throughout the entire training. The FNO architecture does not include the incremental algorithm and resolution. **Bottom: full iFNO architecture.** Our model takes as input functions at different resolutions (discretizations). The operator consists of a lifting layer, followed by a series of iFNO blocks. The loss is used by our method to dynamically update the input resolution and the number of modes $K$ in the Spectral Convolutions. The incremental algorithm is detailed in section 4 and algorithm 1.

function spaces. It approximates the action of the solution operator on any input function, allowing it to generalize across different discretizations and resolutions.

Unlike conventional neural networks, FNO possesses a property termed *discretization-convergence*, meaning it can output predictions at different resolutions, and those predictions converge to a unique solution upon mesh refinement. The discretization-convergence property relies on kernel integration, which, in the case of FNO, is realized using the Fourier transform. A series of such spectral layers, along with normalization layers and non-linear activations, compose the core of the FNO architecture. For many PDE systems, like fluid flows, there is a decay of energy with frequency, i.e., low-frequency components usually have larger magnitudes than high-frequency ones. Therefore, as an inductive bias and a regularization procedure to avoid overfitting, FNO consists of a frequency truncation function $T_K$ in each layer that only allows the lowest $K$ Fourier modes to propagate to the next layer, and the other modes are zeroed out, as shown in Figure 1.

It is thus crucial to select the correct number of frequency modes $K$ in various spectral layers in FNO to achieve optimal generalization performance. Too few number of modes leads to underfitting, while too many can result in overfitting. Further, we need to train on high-enough resolution data to capture all the physical dynamics and approximate the ground-truth solution operator faithfully.

**Our approach:** We propose the Incremental Fourier neural operator (iFNO) to address the above challenges. iFNO incrementally and automatically increases the number of frequency modes $K$ and resolution $R$ during training. It leverages the explained ratio as a metric to dynamically increase the number of frequency modes. The explained ratio characterizes the amount of information in the underlying spectrum that the current set of modes can explain. A small explained ratio (i.e., below a fixed threshold) indicates that the current modes are insufficient to represent the target spectrum and that more high-frequency modes should be added, as illustrated in Figure 1. Simultaneously, iFNO also starts by first training on low-resolution data and progressively increases the resolution, akin to the principle of curriculum learning, thereby increasing training efficiency.

**Our contributions are summarized as follows:**

1. We introduce the Incremental Fourier Neural Operator (iFNO), a novel approach that progressively increases the number of spectral coefficients and the data resolution used during training. These can be applied jointly or individually.

2. We perform a thorough empirical validation across a range of partial differential equation (PDE) problems, including Burgers, Darcy Flow, the Navier-Stokes Equation, and the Kolmogorov Flow.

3. Using our proposed iFNO, we demonstrate upto a 38% lower testing error by acting as a dynamical spectral regularization scheme, using 20% fewer frequency modes compared to the existing FNO, while also achieving a 46% faster performance and a 2.8x reduction in model size, enabling larger scale simulations.

## 2 Related Works

The applications of neural networks on partial differential equations have a rich history (Lagaris et al., 1998; Dissanayake & Phan-Thien, 1994). These deep learning methods can be classified into three categories: (1) ML-enhanced numerical solvers such as learned finite element, finite difference, and multigrid solvers (Kochkov et al., 2021; Pathak et al., 2021; Greenfeld et al., 2019) ; (2) Network network-based solvers such as Physics-Informed Neural Networks (PINNs), Deep Galerkin Method, and Deep Ritz Method (Raissi et al., 2019; Sirignano & Spiliopoulos, 2018; Weinan & Yu, 2018); and (3) the data-driven surrogate models such as (Guo et al., 2016; Zhu & Zabaras, 2018; Bhatnagar et al., 2019). Among them, the machine learning surrogate models directly parameterized the target mapping based on the dataset. They do not require a priori knowledge of the governing system and usually enjoy a light-fast inference speed. Recently, a novel class of surrogate models called neural operators was developed (Li et al., b; Lu et al., 2021; Kissas et al., 2022). Neural operators parameterize the PDEs' solution operator in the function spaces and leverage its mathematical structures in their architectures. Consequentially, neural operators usually have a better empirical performance (Takamoto et al., 2022) and theoretical guarantees Kovachki et al. (2021) combined with conventional deep learning models.

The concept of implicit spectral bias was first proposed by Rahaman et al. as a possible explanation for the generalization capabilities of deep neural networks. There have been various results towards theoretically understanding this phenomenon Cao et al.; Basri et al.. Fridovich-Keil et al. further propose methodologies to measure the implicit spectral bias on practical image classification tasks. Despite a wealth of theoretical results and empirical observations, there has been no prior research connecting the implicit spectral bias with FNO to explain its good generalization across different resolutions.

The notion of incremental learning has previously been applied to the training of PINNs Krishnapriyan et al. (2021); Huang & Alkhalifah. However, these focus on a single PDE instance and apply incremental learning to the complexity of the underlying PDE, e.g., starting with low-frequency wavefields first and using high-frequency ones later Huang & Alkhalifah. Our work is orthogonal to this direction, and rather than modifying the underlying PDE, we directly incrementally increase the model's capacity and the data's resolution.

## 3 Fourier Neural Operator

Fourier Neural Operators belong to the family of neural operators, which are formulated as a generalization of standard deep neural networks to operator setting (Li et al., b). A neural operator learns a mapping between two infinite dimensional spaces from a finite collection of observed input-output pairs. Let $D$ be a bounded, open set and $\mathcal{A}$ and $\mathcal{U}$ be separable Banach spaces of functions of inputs and outputs.

We want to learn a neural operator $\mathcal{G}_\theta : \mathcal{A} \times \theta \to \mathcal{U}$ that maps any initial condition $a \in \mathcal{A}$ to its solution $u \in \mathcal{U}$. The neural operator $\mathcal{G}_\theta$ composes linear integral operator $\mathcal{K}$ with pointwise non-linear activation function $\sigma$ to approximate highly non-linear operators.

**Definition 3.1** (Neural operator). The neural operator $\mathcal{G}_\theta$ is defined as follows:

$$\mathcal{G}_\theta := \mathcal{Q} \circ (W_L + \mathcal{K}_L) \circ \cdots \circ \sigma (W_1 + \mathcal{K}_1) \circ \mathcal{P},$$

where $\mathcal{P}, \mathcal{Q}$ are the pointwise neural networks that encode the lower dimension function into higher dimensional space and decode the higher dimension function back to the lower dimensional space. The model stack $L$ layers of $\sigma (W_l + \mathcal{K}_l)$ where $W_l$ are pointwise linear operators (matrices), $\mathcal{K}_l$ are integral kernel operators, and $\sigma$ are fixed activation functions. The parameters $\theta$ consist of all the parameters in $\mathcal{P}, \mathcal{Q}, W_l, \mathcal{K}_l$.

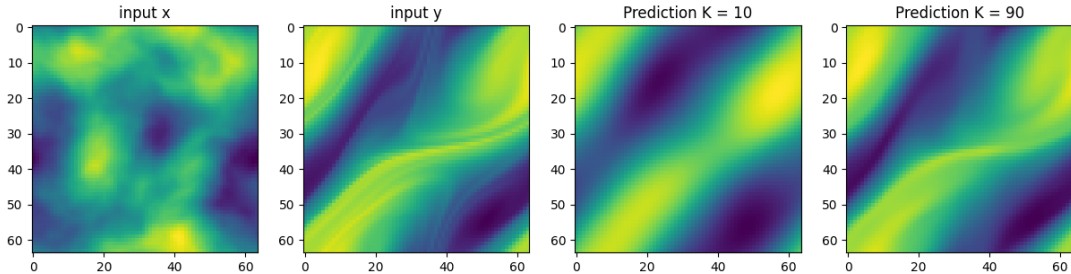

Figure 2: FNO with higher frequency modes captures smaller-scale structures in the fluid. Prediction of the Darcy flow by FNO with $K = 10$ and $K = 90$. Insufficient modes lead to overly strong dumping and fail to capture the finer details in Darcy flow.

Li et al. (a) proposes FNO that adopts a convolution operator for $\mathcal{K}$ as shown in Figure 1, which obtains state-of-the-art results for solving PDE problems.

**Definition 3.2** (Fourier convolution operator). Define the Fourier convolution operator $\mathcal{K}$ as follows:

$$(\mathcal{K}v_t)(x) = \mathcal{F}^{-1}\left(R \cdot T_K\left(\mathcal{F}v_t\right)\right)(x) \quad \forall x \in D,$$

where $\mathcal{F}$ and $\mathcal{F}^{-1}$ are the Fourier transform and its inverse, $R$ is a learnable transformation and $T_K$ is a fixed truncation that restricts the input to lowest $K$ Fourier modes.

FNO is discretization-convergent, such that the model can produce a consistent solution for any query points, potentially not in the training grid. In other words, FNO can be trained on low resolution and evaluated at high resolution. This property is highly desirable as it allows a transfer of solutions between different grid resolutions and discretizations.

**Frequency truncation.** To ensure discretization-convergence, FNO truncates the Fourier series $\hat{\mathcal{F}}v$ at a maximal number of modes $K$ using $T_K$. In this case, for any discrete frequency mode $k \in \{1, ..., K\}$, we have $\hat{\mathcal{F}}v(k) \in \mathbb{C}^{\mathrm{C}}$ and $R(k) \in \mathbb{C}^{\mathrm{C} \times \mathrm{C}}$, where C is the channel dimension of the input $v$. The size of linear parameterization $R$ depends on $K$. For example, $R$ has the shape of $K^d \times \mathrm{C} \times \mathrm{C}$ for a $d$-dimensional problem. We also denote the modes $1, ..., K$ as the effective frequency modes. In the standard FNO, Li et al. (a) view $K$ as an additional hyperparameter to be tuned for each problem.

**Frequency strength.** As $R(k)$ reflects how the frequency mode $k$ is transformed, we can measure the strength of transforming the frequency mode $k$ by the square of the Frobenius norm of $R(k)$, such that:

$$S_k = \sum_i^{\mathrm{C}} \sum_j^{\mathrm{C}} |R_{k,i,j}|^2, \tag{1}$$

where $S_k$ denotes the strength of the $k$-th frequency mode. A smaller $S_k$ indicates that the $k$-th frequency mode is less important for the output.

**Implicit spectral bias in neural networks** It has been well studied that neural networks implicitly learn low-frequency components first, and then learn high-frequency components in a later stage Rahaman et al.; Xu et al. (2019). This phenomenon is known as implicit spectral bias, and it helps explain the excellent generalization ability of overparameterized neural networks.

Since FNO performs linear transformation $R$ in the frequency domain, the frequency strength $S_k$ of each frequency mode $i$ is directly related to the spectrum of the resulting model. As FNO is a neural network and trained by first-order learning algorithms, it follows the implicit spectral bias such that the lower frequency modes have larger strength in $R$. This explains why FNO chooses to preserve a set containing the lowest frequency modes, instead of any arbitrary subset of frequencies.

**Low-frequency components in PDEs** Learning frequency modes is important as large-scale, low-frequency components usually have larger magnitudes than small-scale, high-frequency components in PDEs. For dissipative systems (with diffusion terms) such as the viscous Burgers' equation and incompressible Navier-Stokes equation, the energy cascade involves the transfer of energy from large scales of motion to the small scales, which leads to the Kolmogorov spectrum with the slope of $k^{-5/3}$ in the inverse cascade range (Figure 3), and $k^{-3}$ in the direct-cascade range (Boffetta et al., 2012). The smallest scales in turbulent flow is called the Kolmogorov microscales. Therefore, one should choose the model frequencies with respect to the underlying equation frequencies when designing machine learning models. It would be a challenge to select the correct model frequencies in advance, without knowing the properties of the underlying PDEs.

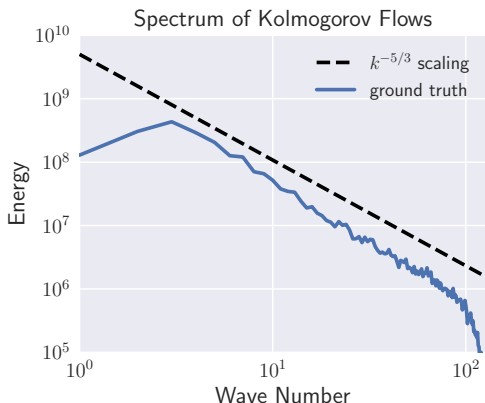

Figure 3: The spectrum of Kolmogorov flow decays exponentially with the Kolmogorov scale of 5/3 in the inverse cascade range.

---

**Algorithm 1** *Incremental Learning of FNO*

---

**Input:** Initial modes $K_0$, mode buffers $b$, threshold $\alpha$, epoch $e$ to increase resolution, downsample factors $r$, data resolution $R_{data}$
**Initialize:** Randomly initialize $R_0 \in \mathbb{C}^{(K_0+b)\times C\times C}$, set initial resolution $R_{curr} = R_{data}/r_1$
**For iteration** $t$ **in** $1,...,T$**:**
    compute $s_t = [S_1, S_2, ..., S_{K_{t-1}+b}]$ {compute the frequency strength for each mode}
    $K_t \leftarrow K_{t-1}$
    **While** $g(K_t, s_t) < \alpha$**:** {find $K_t$ that explains at least $\alpha$ of $s_t$}
        $K_t \leftarrow K_t + 1$
    construct $R_t \in \mathbb{C}^{(K_t+b)\times C\times C}$ where $R_t[0:(K_{t-1}+b),:,:] = R_{t-1}$ and initialize the rest randomly
    **If** ($t \bmod e = 0$)
        Set next resolution $R_{curr} = R_{data}/r_{i+1}$
    **Train FNO model with** $(K_t, R_{curr})$

---

## 4 Incremental Fourier Neural Operator

Although FNO has shown impressive performance in solving PDEs, the training remains challenging. In this section, we discuss the main difficulties in training FNOs and propose the Incremental Fourier Neural Operator (iFNO) to address the challenges.

### 4.1 Incremental frequency FNO

While frequency truncation $T_K$ ensures the discretization-invariance property of FNO, it is still challenging to select an appropriate number of effective modes $K$, as it is task-dependent and requires careful hyperparameter tuning.

Inappropriate selection of $K$ can lead to severe performance degradation. Setting $K$ too small will result in too few frequency modes, such that it doesn't approximate the solution operator well, resulting in underfitting. As shown in Figure 2, in the prediction of Darcy flow, FNO requires higher frequency modes to capture small-scale structures in the fluid for better performance. On the other hand, a larger $K$ with too many effective frequency modes may encourage FNO to interpolate the noise in the high-frequency components, leading to overfitting.

Previously, people have chosen the number of effective modes in FNO by estimating the underlying data frequencies or using heuristics borrowed from pseudo-spectral solvers such as the 2/3 dealiaising rule (Hou & Li, 2007), i.e., picking the max Fourier modes as 2/3 of the training data resolution. However, it's usually not easy to estimate the underlying frequency, and the resolution of data may vary during training and testing.

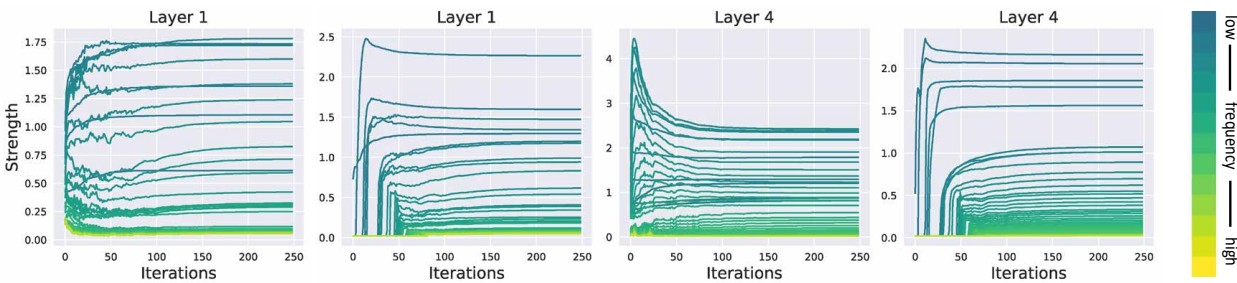

Figure 4: Frequency evolution of first and fourth Fourier convolution operators in FNO and iFNO during the training on Burgers' equation. We visualize FNO on the left figure and iFNO on the right figure for each layer. Each frequency strength $S_k$ is visualized across training. FNO is tuned with the optimal weight decay strength.

These problems lead to hand-tuning or grid-searching the optimal effective modes, which can be expensive and wasteful. We propose the *incremental Fourier Neural Operator (iFNO)* that starts with a "small" FNO model with limited low-frequency modes and gradually increases $K$ and resolution $R$ based on the training progress or frequency evolution. One of the key challenges in training iFNO is determining an appropriate time to proceed with the increase of $K$. We consider two variants of iFNO (method), which means we only increase the frequency modes based on the method.

**iFNO (Loss):** A common practice is to use the training progress as a proxy to determine the time, as seen in previous work such as Liu et al.. Specifically, we let the algorithm increase $K$ adaptively only when there is a decrease in training loss between $N$ consecutive epochs that is lower than a threshold $\epsilon$. We denote it as the iFNO (loss) algorithm. Despite its simplicity, this heuristic has shown to be effective in practice. However, finding an appropriate $\epsilon$ and $N$ is challenging as it depends on the specific problem and the loss function. To address this issue, we propose a novel method to determine the expanding time by directly considering the frequency evolution in the parameter space.

**iFNO (Freq):** As shown in Equation 1, $i$-th frequency strength $S_i$ reflects the importance of the $i$-th frequency mode in the model. The entire spectrum of the transformation $R$ can be represented by the collection of $p$ frequency strengths (when $p$ is sufficiently large):

$$s_p = [S_1, S_2, ..., S_p], \tag{2}$$

where $s_p \in \mathbb{R}^p$. When applying the frequency truncation $T_K$ $(K < p)$, we can measure how much the lowest $K$ frequency modes explain the total spectrum by computing the *explanation ratio* $g(K, s_p)$.

$$g(K, s_p) = \frac{\sum_{k=1}^{K} S_k}{\sum_{k=1}^{p} S_k}. \tag{3}$$

We define a threshold $\alpha$, which is used to determine if the current modes can well explain the underlying spectrum. If $g(K, s_p) > \alpha$, it indicates that the current modes are sufficient to capture the important information in the spectrum, and thus no additional modes need to be included in FNO. Otherwise, more frequency modes will be added into the model until $g(K, s_p) > \alpha$ is satisfied.

Although $s_p$ reflects the entire spectrum when $p$ is sufficiently large, maintaining a large $p$ is unnecessary and computationally expensive. Instead, we only maintain a truncated spectrum $s_{K+b} \in \mathbb{R}^{K+b}$, which consists of $K$ effective modes and $b$ buffer modes. The buffer modes contain all mode candidacies that potentially would be included as effective modes in the later stage of training.

In practice, at iteration $t$ with $K_t$ effective modes, we construct a transformation $R_t \in \mathbb{C}^{(K_t+b) \times C \times C}$ for 1D problems. After updating $R_t$ at iteration $t + 1$, we will find $K_{t+1}$ that satisfies $g(K_{t+1}, s_t) > \alpha$. In this way, we can gradually include more high-frequency modes when their evolution becomes more significant, as illustrated in Figure 1. We denote this variant as iFNO (Freq). We also describe how iFNO (Freq) is trained for solving 1D problems in Algorithm 1. iFNO is extremely efficient as the training only requires a part of frequency modes. In practice, it also converges to a model requiring fewer modes than the baseline FNO without losing any performance, making the model efficient for inference. Notably, the cost of computing the

frequency strength and explanation ratio is negligible compared to the total training cost. It can be further reduced by performing the determination every $T$ iterations, which does not affect the performance.

## 4.2 Regularization in FNO training

As discussed in the previous section, learning low-frequency modes is essential for the successful training of FNO. However, the standard regularization techniques used in training neural networks are not capable of explicitly promoting the learning of low-frequency modes.

As shown in Figure 4, although FNO with well-tuned weight decay strength successfully regularizes the high-frequency modes, its regularization can be too strong for certain layers (such as the fourth operator in the figure). This leads to the instability of the frequency evolution, which damages the generalization performance, as shown in Table 2. On the other hand, insufficient decay strength cannot regularize the high-frequency modes and can even lead to overfitting the associated high-frequency noise.

**Dynamic Spectral Regularization:** However, we find that iFNO can be served as a dynamic spectral regularization process. As shown in Figure 4, iFNO properly regularizes the frequency evolution without causing instability or overfitting high-frequency modes. As we will present in the experiment section, the dynamic spectral regularization gives iFNO a significant generalization improvement.

## 4.3 Incremental resolution FNO

Finally, we introduce the incremental algorithm on the resolution part of our approach.

**Computational cost in training large-scale FNO** Training the FNO on large-scale, high-resolution data to solve PDEs with high-frequency information, a large effective mode number $K$ is necessary to capture the solution operator. However, this requires constructing a large-scale transformation $R$, which dominates the computational cost of all the operations in FNO. This makes the training and inference more computationally expensive. For example, the Forecastnet (Pathak et al., 2022) for global weather forecast based on FNO is trained on 64 NVIDIA® Tesla® A100 GPUs, only covering a few key variables. To simulate all the weather variables, it potentially needs the parallel FNO, which scales to 768 GPUs (Grady II et al., 2022).

**Incremental Resolution:** To mitigate the computational cost of training FNO on high-resolution data, we incorporate curriculum learning to improve our model (Soviany et al. (2022)). This approach involves training the models in a logical sequence, where easier samples are trained first and gradually progressing to more challenging ones. In our case, we start with lower-resolution images, allowing the model to capture large-scale features first, potentially accelerating learning and reducing computational costs in the early stages of training, and then moving on to higher-resolution images. This approach allows the model to initially focus on learning low-frequency components of the solution using small data, avoiding wasted computation and overfitting. Then, the model progressively adapts to capture higher frequency components as the data resolution increases. This model combines both incremental frequency (specifically the iFNO Freq)) and incremental resolution and is denoted by iFNO. Algorithm 1 showcases the joint implementation.

Table 1: **Evaluation on Re5000 dataset on full data regime**. The average training L2 loss and testing L2 loss on super-resolution are reported on 3 runs with their standard deviation.

| Method | Train L2 | Test L2 (Super-Res) (1e-1) | Training Time (Mins) |
|---|---|---|---|
| iFNO | **0.149 ± 0.004** | 0.961 ± 0.022 | **670 ± 40** |
| Standard FNO | 0.169 ± 0.004 | 0.988 ± 0.031 | 918 ± 10 |
| iFNO (Freq) | 0.163 ± 0.003 | 1.019 ± 0.031 | 700 ± 30 |
| iFNO (Loss) | 0.155 ± 0.005 | **0.948 ± 0.040** | 690 ± 30 |
| iFNO (Resolution only) | 0.155 ± 0.003 | 0.974 ± 0.010 | 668 ± 40 |

Table 2: **Evaluation on various PDEs in the low-data regime**. The average training and testing L2 loss are reported with their standard deviation over 3 repeated runs. All iFNO variants are compared with standard FNO with fixed $K = 10, 30, 60, 90$ number of modes. Losses have been divided by the scale under the column.

| Method | Burgers | | Darcy Flow | | Navier Stokes (FNO2D) | |
|---|---|---|---|---|---|---|
| | Train (1e − 3) | Test (1e − 2) | Train (1e − 3) | Test (1e-2) | Train (1e − 2) | Test (1) |
| iFNO | **0.418 ± 0.002** | **0.139 ± 0.006** | 2.466 ± 0.855 | 0.129 ± 0.028 | 3.533 ± 0.003 | 0.106 ± 0.011 |
| iFNO (Freq) | 0.422 ± 0.001 | 0.141 ± 0.001 | 1.491 ± 0.298 | 0.215 ± 0.010 | **1.548 ± 0.029** | 0.177 ± 0.008 |
| iFNO (Loss) | 0.430 ± 0.001 | 0.140 ± 0.007 | 2.262 ± 0.985 | 0.198 ± 0.029 | 2.466 ± 0.092 | 0.202 ± 0.020 |
| iFNO (Res) | 0.423 ± 0.001 | 0.146 ± 0.006 | **1.410 ± 0.020** | **0.123 ± 0.001** | 3.576 ± 0.021 | **0.105 ± 0.001** |
| FNO (10) | 1.135 ± 0.002 | 0.270 ± 0.006 | 1.471 ± 0.223 | 0.162 ± 0.003 | 5.133 ± 0.182 | 0.166 ± 0.027 |
| FNO (30) | 0.455 ± 0.001 | 0.156 ± 0.002 | 1.462 ± 0.102 | 0.140 ± 0.011 | 3.262 ± 0.001 | 0.154 ± 0.001 |
| FNO (60) | 0.421 ± 0.001 | 0.153 ± 0.006 | 1.468 ± 0.212 | 0.130 ± 0.007 | - ± - | − ± − |
| FNO (90) | 0.423 ± 0.001 | 0.146 ± 0.006 | 1.425 ± 0.212 | 0.127 ± 0.004 | − ± − | − ± − |

| Method | Navier Stokes (FNO3D) | | Kolmogorov Flow | |
|---|---|---|---|---|
| | Train (1e − 2) | Test (1e − 1) | Train (1) | Test (1) |
| iFNO | 1.694 ± 0.009 | 0.308 ± 0.018 | 0.832 ± 0.279 | 0.358 ± 0.009 |
| iFNO (Freq) | 1.402 ± 0.001 | 0.559 ± 0.011 | 1.455 ± 0.008 | 0.487 ± 0.004 |
| iFNO (Loss) | **0.832 ± 0.001** | **0.281 ± 0.011** | 0.740 ± 0.230 | 0.381± 0.010 |
| iFNO (Res) | 1.704 ± 0.032 | 0.519 ± 0.016 | 0.882 ± 0.067 | **0.342± 0.001** |
| FNO (10) | 1.892 ± 0.020 | 0.527 ± 0.002 | 2.111 ± 0.009 | 0.543 ± 0.009 |
| FNO (30) | 1.298 ± 0.001 | 0.544 ± 0.001 | 1.053 ± 0.190 | 0.384 ± 0.010 |
| FNO (60) | − | − | 0.562 ± 0.001 | 0.348 ± 0.001 |
| FNO (90) | − | − | **0.475 ± 0.001** | 0.343 ± 0.001 |

## 5 Evaluations

In this section, we detail the experimental setting, datasets used and analyze the empirical results.

### 5.1 Experimental Setup

We evaluate iFNO and its variants on five different datasets of increasing difficulty. We retain the structure of the original FNO, which consists of stacking four Fourier convolution operator layers with the ReLU activation and batch normalization. More specifically, for the time-dependent problems, we use an RNN structure in time that directly learns in space-time[1]. The initial and coefficient conditions are sampled from Gaussian random fields (Nelsen & Stuart, 2021) for all the datasets.

**Burgers' Equation**  We consider the 1D Burgers' equation, which is a non-linear PDE with various applications, including modeling the flow of a viscous fluid. The 1D Burgers' equation takes the form:

$$\partial_t u(x,t) + \partial_x \left( u^2(x,t)/2 \right) = \nu \partial_{xx} u(x,t), \quad t \in (0,1]$$
$$u(x,0) = u_0(x), \qquad x \in (0,1)$$

where $u_0$ is the initial condition and $\nu$ is the viscosity coefficient. We aim to learn the operator $\mathcal{G}_\theta$, mapping the initial condition to the solution. The training is performed under the resolution of 1024, and we consider the learning rate halved every 50 epochs and weight decay to be 0.0001.

---

[1]We share our code via the neural operator codebase.

**Darcy Flow**  Next, we consider the steady-state of the 2D Darcy Flow on the unit box, which is the second-order, linear, elliptic PDE:

$$-\nabla \cdot (a(x)\nabla u(x)) = f(x), \quad x \in (0,1)^2$$
$$u(x) = 0, \qquad x \in \partial(0,1)^2$$

With a Dirichlet boundary where $a$ is the diffusion coefficient, and $f = 1$ is the forcing function. We are interested in learning the operator mapping the diffusion coefficient to the solution. The resolution is set to be $421 \times 421$.

**Navier-Stokes Equation**  We also consider the 2D+time Navier-Stokes equation for a viscous, incompressible fluid on the unit torus:

$$\partial_t w(x,t) + u(x,t) \cdot \nabla w(x,t) = \nu \Delta w(x,t) + f(x),$$
$$\nabla \cdot u(x,t) = 0, \ \ x \in (0,1)^2$$
$$w(x,0) = w_0(x), \ \ t \in (0,T]$$

where $w = \nabla \times u$ is the vorticity, $w_0$ is the initial vorticity, $\nu \in \mathbb{R}_+$ is the viscosity coefficient, and $f(x) = 0.1\big(\sin(2\pi(x_1 + x_2)) + \cos(2\pi(x_1 + x_2))\big)$ is the forcing function. We are interested in learning the operator mapping the vorticity up to time 10 to the vorticity up to some later time. We experiment with the hardest viscosity task, i.e., $\nu = 1e-5$. The equivalent Reynolds number is around 500 normalized by forcing and domain size. We set the final time $T = 20$ as the dynamic becomes chaotic. The resolution is fixed to be $64 \times 64$.

We consider solving this equation using the 2D FNO with an RNN structure in time or using the 3D FNO to do space-time convolution, similar to the benchmark in Li et al. (a). As 3D FNO requires significantly more parameters, evaluating 3D FNO with more than 30 modes is computationally intractable due to the hardware limitation.

**Kolmogorov Flow:**  We evaluate the models on a very challenging dataset of high-frequency Kolmogorov Flow. The Re5000 problem is the most challenging dataset we consider. It is governed by the 2D Navier-Stokes equation, with forcing $f(x) = \sin(nx_2)\hat{x}_1$, where $x_1$ is the unit vector and a larger domain size $[0, 2\pi]$, as studied in (Li et al., 2021; Kochkov et al., 2021). More specifically, the Kolmogorov flow (a form of the Navier-Stokes equations) for a viscous, incompressible fluid,

$$\frac{\partial u}{\partial t} = -u \cdot \nabla u - \nabla p + \frac{1}{Re}\Delta u + \sin(ny)\hat{x}, \quad \nabla \cdot u = 0,$$
$$\text{on } [0, 2\pi]^2 \times (0, \infty)$$

with initial condition $u(\cdot, 0) = u_0$ where $u$ denotes the velocity, $p$ the pressure, and $Re > 0$ is the Reynolds number. It has an extremely high Reynolds number with $Re = 5000$. The resolution of this dataset is $128 \times 128$. We consider learning the evolution operator from the previous time step to the next based on 2D FNO. We also consider a variation of the dataset with a lower $Re$ value, which we call the Kolmogorov Flow problem. This resolution is fixed to be $256 \times 256$ to show high-frequency details.

**Methods**  We present our results for each dataset where we consider four standard FNO baselines with different numbers of frequency modes $K = 10, 30, 60, 90$. For each dataset, all methods are trained with the same hyperparameters, including the learning rate, weight decay, and the number of epochs. Unless otherwise specified, we use the Adam optimizer to train for 500 epochs with an initial learning rate of 0.001, which is halved every 100 epochs, and weight decay is set to be 0.0005. For the re5000 dataset, we train for 50 epochs, and experiment details are explained in the Appendix. All experiments are run using NVIDIA® Tesla® V100/A100 GPUs. We also test our incremental methods on Tensorized Factorized Neural Operators Kossaifi et al. (2023) by sweeping over different factorization schemes and ranks for the weights.

**Hyperparameter settings** We find iFNO is *less sensitive* to the choice of hyperparameters across different datasets. This property allows the same iFNO architecture and hyperparameters to be directly applied to a wide range of problems without expensive parameter validation and without needing to set an appropriate $K$. Consequently, we use the same hyperparameters for all datasets, including the number of initial modes $K_0 = 1$, the number of buffer modes $b = 5$, and the threshold $\alpha = 0.9999$. We also use a uniform threshold $\epsilon = 0.001$ in iFNO (Loss).

## 5.2 Experiment details for Re5000 dataset - iFNO

Regarding setting the scheduler, the incremental resolution approach significantly alters the loss landscape and distribution of the training data each time the resolution is increased. As noted by Smith (2017), cyclic learning rates oscillating between higher and lower values during training can be highly beneficial when dealing with such changing loss landscapes. Therefore, our incremental resolution training methodology adopts a combination of the cyclic learning rate and step learning rate scheme. We see that this combination provides improved and more stable convergence throughout the curriculum of incremental resolution increases, contributing to strong empirical results for both the incremental resolution model and the iFNO model. A triangular cyclic learning rate allows the model to first explore with larger learning rates to find new optimal parameters after a resolution increase before narrowing down with smaller rates to converge. This prevents the model from getting stuck in sub-optimal solutions. Our experiments utilize 3 different modes, i.e., triangular, triangular with decreased height, and exponential decay. We also sweep over various step sizes and maximum and minimum bounds. The cyclic period is set to $e$ epochs. Although the cyclicLR method is indeed suitable, we find that after we switch to the highest resolution, using stepLR converges to a better minimum compared to cyclicLR. This makes sense, as once we start training on the highest resolution, instead of changing our learning rates in an oscillatory manner, we should stabilize and take smaller steps for convergence, like how the baseline FNO is trained.

Our algorithm increases resolution every $e$ epoch from 32 to 64 and then 128. We use powers of 2 for optimal Fast Fourier Transform speed. The hyperparameter $e$ can be tuned, and $r$ is a list determining how low we sub-sample data. In the re5000 dataset case, we have set $e = 10$ and $r = [4, 2, 1]$, which results in starting from resolution 32 all the way up to the highest resolution 128.

## 5.3 Results

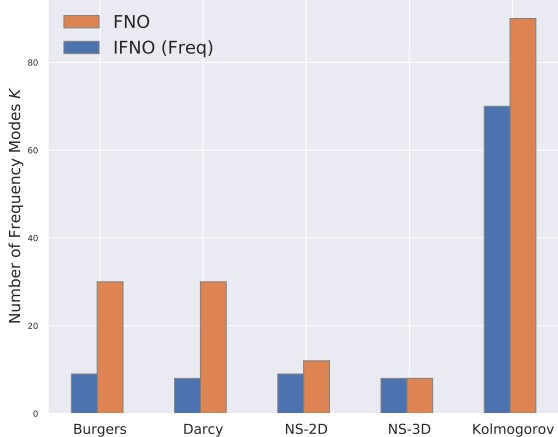

(a) Number of frequency modes $K$ in the converged FNO and iFNO models across datasets. We report $K$ in the first Fourier convolution operator. NS denotes Navier-Stokes equations.

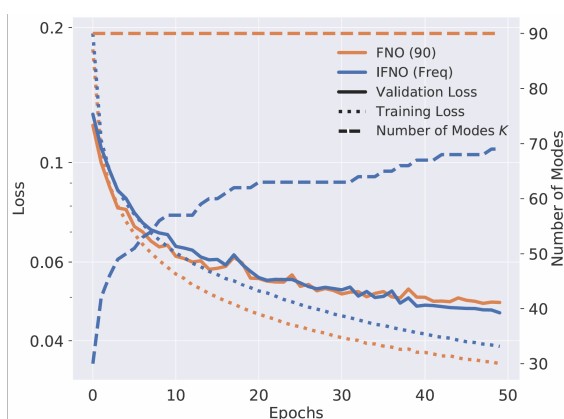

(b) Testing loss, training loss, and number of modes $K$ during the training of FNO and iFNO on Kolmogorov flow.

Figure 5: iFNO comparison, both modes and loss curves.

Table 3: Evaluation on various PDEs in the full data regime. The average training and testing L2 loss are reported with their standard deviation over 3 repeated runs. Frequency-based and loss-based iFNO are compared with standard FNO with fixed $K = 10, 30, 60, 90$ number of modes. Losses have been divided by the scale under the column.

| Method | Burgers | | Darcy Flow | | Navier Stokes (FNO 2D) | |
|---|---|---|---|---|---|---|
| | Train $(1e-3)$ | Test $(1e-3)$ | Train $(1e-3)$ | Test (1e-3) | Train $(1e-1)$ | Test (1e-1) |
| iFNO | **0.891 ± 0.022** | **0.760 ± 0.039** | 4.580 ± 0.855 | **0.264 ± 0.020** | **1.022 ± 0.043** | 0.824 ± 0.016 |
| iFNO (Freq) | 0.898 ± 0.022 | 0.778 ± 0.005 | **2.453 ± 0.260** | 0.387 ± 0.009 | 1.102 ± 0.132 | 1.265 ± 0.044 |
| iFNO (Loss) | 0.919 ± 0.033 | 0.779 ± 0.034 | 3.113 ± 0.887 | 0.322 ± 0.068 | 1.135 ± 0.003 | 1.375 ± 0.224 |
| iFNO (Res) | 0.979 ± 0.053 | 0.854 ± 0.034 | 4.463 ± 0.920 | 0.271 ± 0.010 | 1.112 ± 0.080 | **0.737 ± 0.227** |
| FNO (10) | 2.523 ± 0.028 | 1.063 ± 0.025 | 2.991 ± 0.302 | 0.481 ± 0.047 | 10.66 ± 0.736 | 1.286 ± 0.178 |
| FNO (30) | 1.032 ± 0.017 | 0.868 ± 0.033 | 2.981 ± 0.046 | 0.412 ± 0.029 | 1.223 ± 0.045 | 1.184 ± 0.107 |
| FNO (60) | 0.966 ± 0.023 | 0.845 ± 0.016 | 2.596 ± 0.433 | 0.466 ± 0.013 | 1.112 ± 0.029 | 1.178 ± 0.047 |
| FNO (90) | 0.973 ± 0.042 | 0.851 ± 0.029 | **2.521 ± 0.046** | 0.379 ± 0.043 | 1.155 ± 0.001 | 1.577 ± 0.002 |

| Method | Navier Stokes (FNO 3D) | | Kolmogorov Flow | |
|---|---|---|---|---|
| | Train $(1e-2)$ | Test $(1e-1)$ | Train $(1e-2)$ | Test $(1e-1)$ |
| iFNO | 0.820 ± 0.041 | 0.272 ± 0.003 | 2.665 ± 0.034 | 1.507 ± 0.009 |
| iFNO (Freq) | 0.855 ± 0.007 | **0.262 ± 0.002** | 4.345 ± 0.001 | 1.443 ± 0.001 |
| iFNO (Loss) | 0.832 ± 0.001 | 0.281 ± 0.011 | 3.393 ± 0.001 | 2.684 ± 0.003 |
| iFNO (Res) | **7.533 ± 0.006** | 0.281 ± 0.001 | 2.336 ± 0.003 | **0.985 ± 0.001** |
| FNO (10) | 1.035 ± 0.027 | 0.296 ± 0.016 | 5.462 ± 0.004 | 2.227 ± 0.012 |
| FNO (30) | 0.977 ± 0.001 | 0.587 ± 0.001 | 3.571 ± 0.002 | 1.312 ± 0.012 |
| FNO (60) | – | – | 2.751 ± 0.001 | 1.015 ± 0.001 |
| FNO (90) | – | – | **2.330 ± 0.002** | 1.042 ± 0.002 |

First, we study the performance of our model on the large-scale Re5000 dataset, where we use 36000 training samples and 9000 testing samples. The results in Table 1 show that iFNO achieves significantly better generalization performance than just incrementally increasing either frequency or resolution alone. This highlights the advantages of the iFNO while gaining around a 30% efficiency and much better accuracy than the stand-alone Incremental methods. The training time is reduced due to starting with a very small model and input data. There is one important note to mention that, in fact, for the Re5000 dataset, since the problem is highly challenging, i.e., having a high Reynolds number, we need to reach all 128 modes to capture even the high-frequency components.

Next, we evaluate all methods in the low-data regime across different datasets, where the models only have access to a few data samples. This is common in many real-world PDE problems where accurately labeled data is expensive to obtain. In this regime, the ability to generalize well is highly desirable. We consider a small training set (5 to 50 samples) for Burgers, Darcy, and Navier-Stokes datasets. As the data trajectories are already limited in the Kolmogorov dataset, we choose 8x larger time step (sampling rate) for each trajectory, resulting in fewer time frames.

As shown in Table 2, iFNO consistently outperforms the FNO baselines across all datasets, regardless of the number of modes $K$. It also shows that FNO with larger $K$ achieves lower training loss but overfits the training data and performs poorly on the testing set. On the other hand, iFNO (Freq) achieves slightly higher training loss but generalizes well to the testing set, which demonstrates the effectiveness of the dynamic spectral regularization. We also notice that although iFNO (Loss) has significant fluctuations, especially in the training results, it does achieve better generalization on most datasets against the baselines.

### 5.4   Full Data Results

We also evaluate the models with the most data samples available in each dataset. As shown in Table 3, iFNO achieves good performance comparable with standard FNO.

1. **Generalization:** iFNO achieves up to 38% better generalization performance than the baselines, and all iFNO methods beat the baselines in testing error across tasks like Burgers, Darcy, NS2D+time, and NS3D.

2. **Efficiency:** iFNO exhibits remarkable efficiency gains, ranging from 21% to 46% reduction in computational time compared to the baselines, depending on the task.

3. **Model Size:** iFNO achieves substantial model size reductions, with up to 2.8x smaller model size than the baselines, without compromising performance.

### 5.5   Ablation Studies

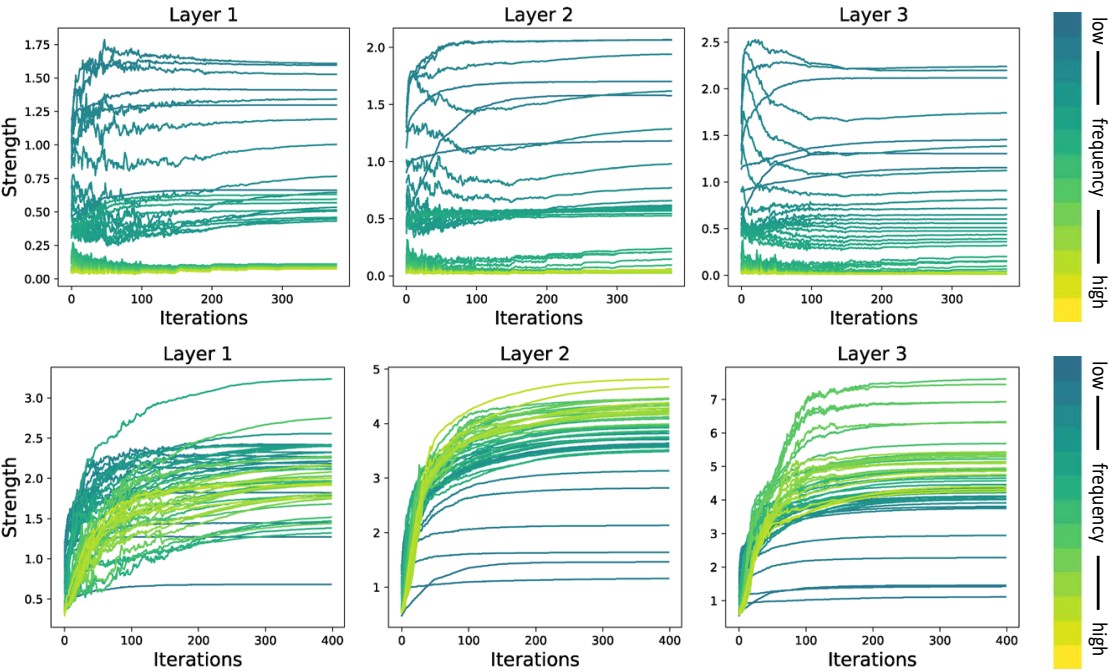

Figure 6:   Frequency evolution of the Fourier convolution operators in FNO with and without weight decay. **Top**: with weight decay; **bottom**: without weight decay. When training with weight decay, the strength of each mode converges around 100 training iterations. And the weights of lower frequency tend to have high strength. On the other hand, the weight continues growing throughout the training without the weight decay. In this case, higher frequencies can also have high strength.

Figure 5b compares FNO (90) and iFNO (Freq) over the training on Kolmogorov flow. It shows that iFNO requires much fewer frequency modes during training. As the linear transformation $R$ dominates the number of parameters and FLOPs in FNO when $K$ is large, iFNO (Freq) can significantly reduce the computational cost and memory usage of training FNO. iFNO achieves better generalization and even requires fewer modes at the end of training. In Figure 5a, we compare iFNO (Freq) with FNO with a particular $K$ that achieves the best performance in each dataset. The results suggest that the trained iFNO (Freq) consistently requires fewer frequency modes than its FNO counterpart, making the iFNO inference more efficient. This also indicates that iFNO can automatically determine the optimal number of frequency modes $K$ during training without predetermining $K$ as an inductive bias that could potentially hurt the performance. To further demonstrate the adaptability and broader applicability of our incremental approach, we conducted a case

study applying iFNO to Tensorized Fourier Neural Operators (TFNOs) Kossaifi et al. (2023). TFNOs represent a recent advancement in efficient operator learning, and we were interested in exploring how our incremental techniques could enhance their performance. The full details of these experiments are presented in Appendix B.6. In summary, we show that TFNOs with our incremental approach (iTFNO) achieve up to 11% better generalization error and up to 22% performance speed up compared to baselines.

## 6 Standard Regularization in training FNO

We also visualize the frequency evolution of FNO and iFNO. As shown in Figure 6, the weight decay used in the optimization has a significant effect on the learned weights. When the weight decay is active, the strength of weights will converge around the first 100 iterations of training. Further, the strength of frequencies is ordered with respect to the wave numbers where the lower frequencies have high strength and the higher frequencies have lower strength. On the other hand, when training without weight decay, the strength of all frequency modes will continuously grow and mix up with each other.

## 7 Conclusion

In this work, we proposed the Incremental Fourier Neural Operator (iFNO) that dynamically selects frequency modes and increases the resolution during training. Our results show that iFNO achieves better generalization while requiring less number of parameters (frequency modes) compared to the standard FNO. We believe iFNO opens many new possibilities, such as solving PDEs with very high resolution and with limited labeled data and training computationally efficient neural operators. In the future, we plan to apply iFNO on more complex PDEs and extend it to the training of physics-informed neural operators.

## 8 Impact statement

This paper presents work with the goal of advancing the field of machine learning and PDEs. FNOs have shown promise to solve challenging physics and engineering problems accurately and efficiently. Our research has made FNOs more optimized through incremental learning and, hence, could lead to FNOs being applied to solve larger real-world problems, allowing for advances in fields like engineering, weather forecasting, and climate science while reducing training times with fewer computing resources. There might also be potential negative societal consequences of our work, but none of them are immediate to be specifically highlighted here.

## 9 Acknowledgements

Robert Joseph George is supported by a Caltech Graduate Fellowship. Anima Anandkumar is supported by the Bren Named Chair, Schmidt AI 2050 Senior fellow, and ONR (MURI grant N00014-18-12624).

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

# A    Additional Results

## A.1    Frequency mode evolution during training

As shown in Figure 7, the effective modes $K$ adaptively increase during the training phase on the Burgers, Darcy, Navier-Stokes (2D & 3D formulation), and Kolmogorov Flows. For smooth problems such as viscous Burgers equation and Darcy Flow, the mode converges in the early epoch, while for problem with high frequency structures such as the Kolmogorov Flows, the mode will continuously increase throughout the training phase.

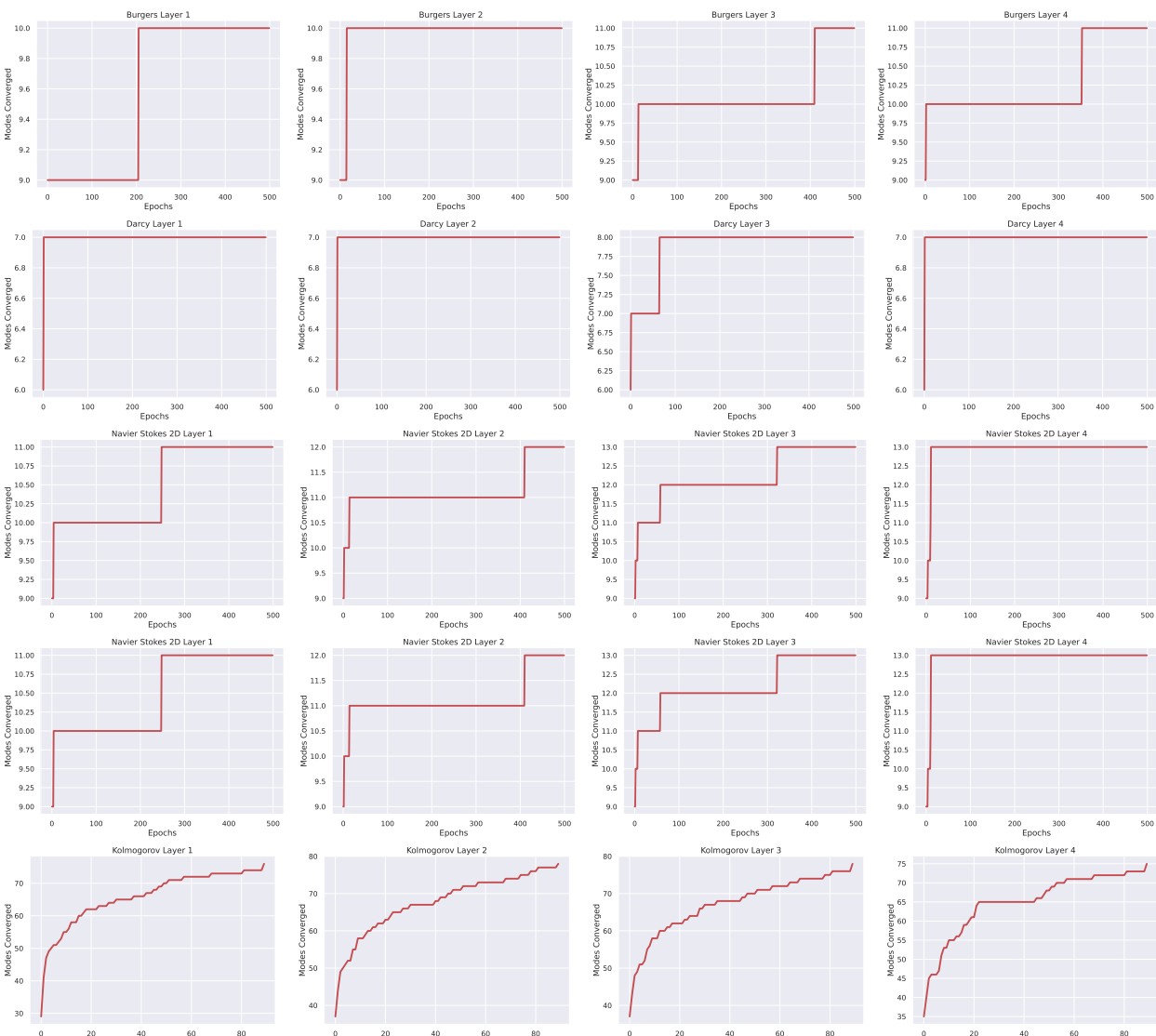

Figure 7: Mode evolution during training for each layer across datasets.

## A.2 Training and testing loss during learning

Figure 8 and 9 show the training curves and testing curves across the five problem settings, ten rows in total. Overall, iFNO methods have a slightly higher training error but lower validation error, indicating its better generalization over the few data regime.

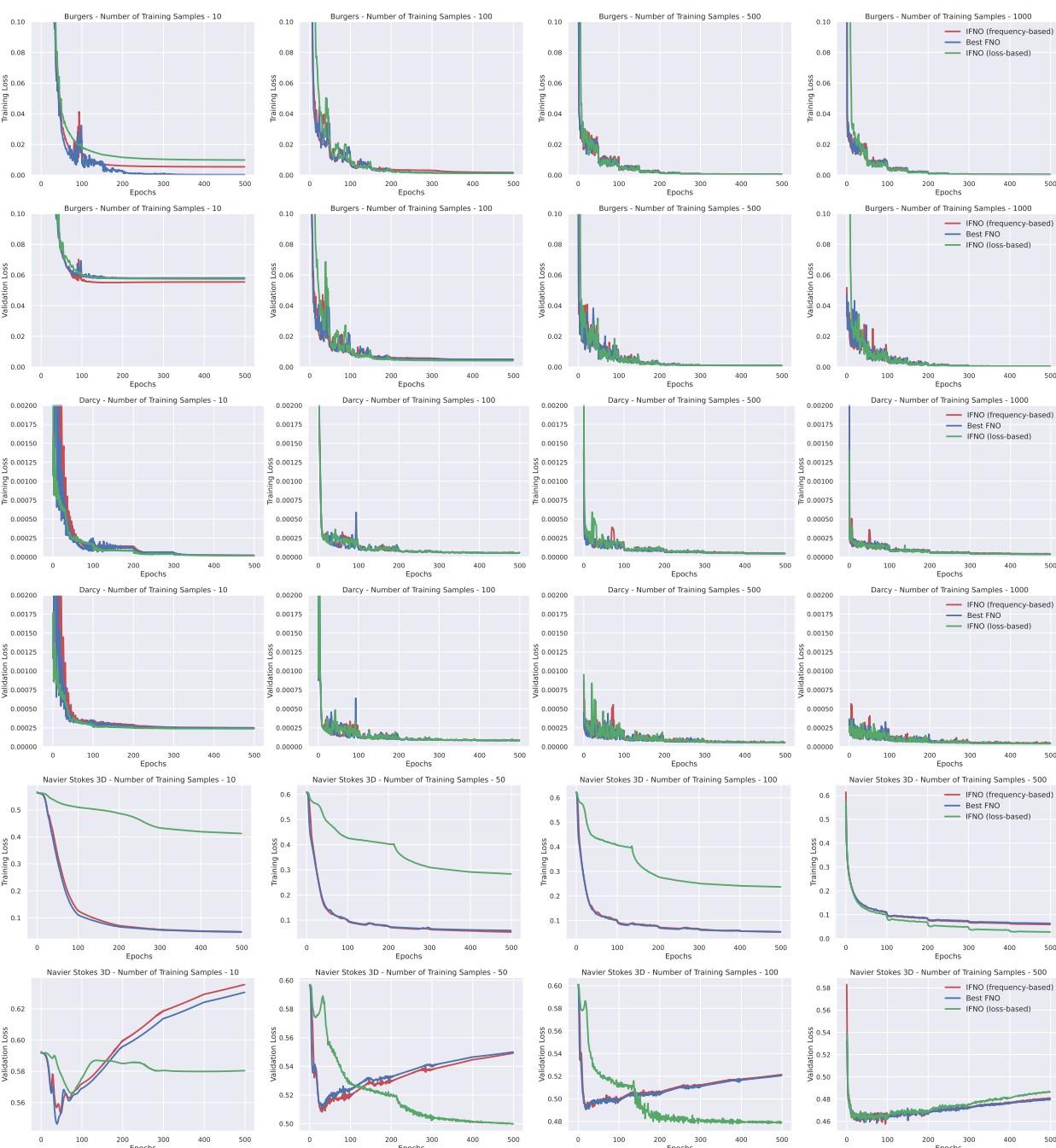

Figure 8: Loss vs Epochs across datasets (part 1: Burgers, Darcy, Navier-Stokes 2D). **Each top row: training L2 loss**. **Each bottom row: validation L2 loss**

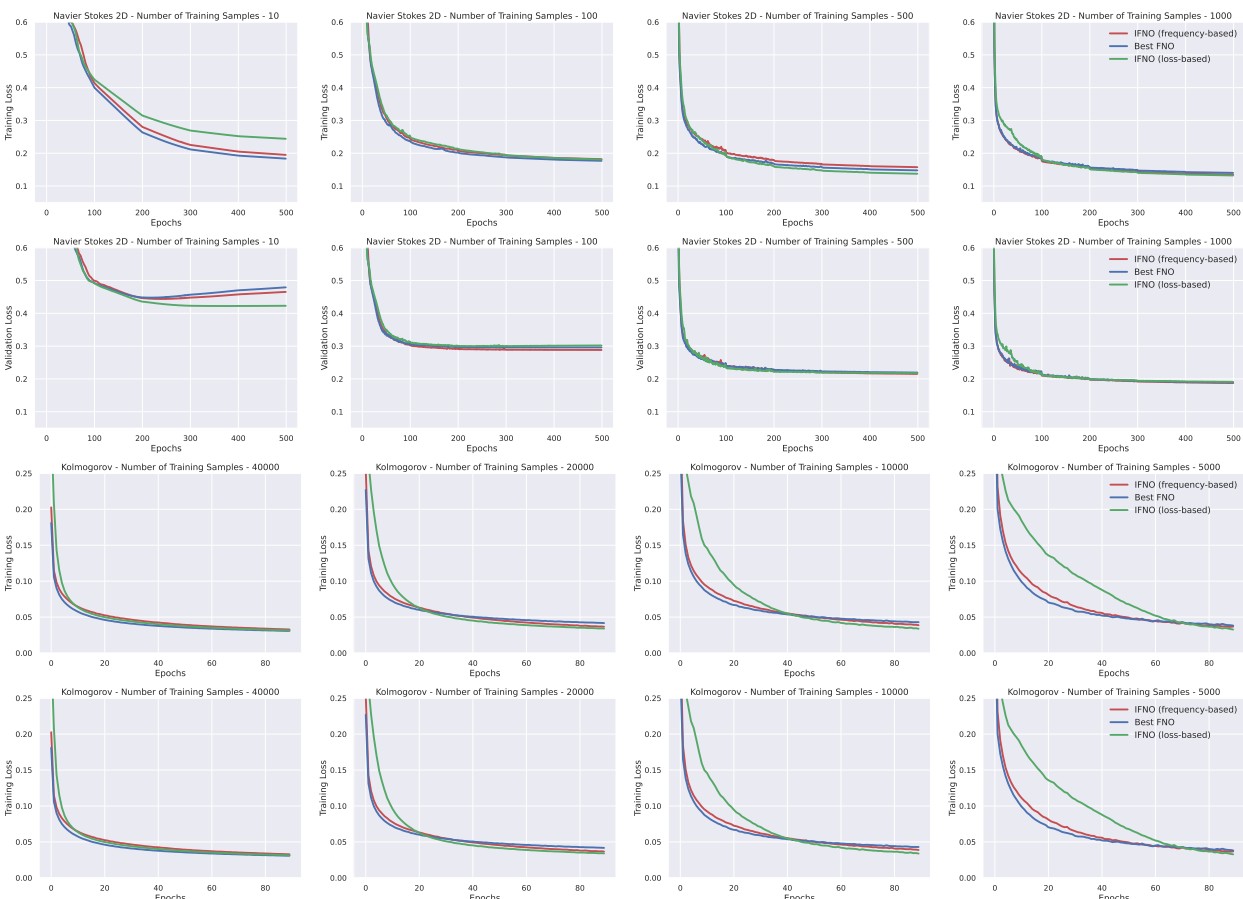

Figure 9: Loss vs Epochs across datasets (part 2: Navier-Stokes 2D, Kolmogorov Flows). **Each top row: training L2 loss**. **Each bottom row: validation L2 loss**

# B  Additional Experiments and Results

## B.1  Selection of Hyperparameters

In Tables 4 and 5 we present the results of our hyperparameter search for  in iFNO (Freq) and $\epsilon$ in iFNO (Loss) respectively, across various tasks. These hyperparameters control the threshold for adding new modes in our incremental approaches. For iFNO (Freq), $\alpha$ represents the cumulative energy threshold in the Explanation Ratio criterion. We explored values ranging from 0.6 to 0.9999, covering a wide spectrum from more aggressive (lower $\alpha$) to more conservative (higher $\alpha$) mode addition strategies. For iFNO (Loss), $\epsilon$ represents the threshold for the relative decrease in loss that triggers the addition of new modes. We investigated values from 0.1 to 0.00001, encompassing both rapid (higher $\epsilon$) and gradual (lower $\epsilon$) mode addition schemes. For most tasks, moderate values of $\alpha$(around 0.99) and $\epsilon$ (around 0.001) tend to perform well, balancing between adding sufficient modes for expressiveness and avoiding overfitting. However, some tasks like Kolmogorov Flow benefit from more aggressive mode addition (lower $\alpha$ or higher $\epsilon$), likely due to the complexity of the underlying dynamics.

Table 4: Evaluation of $\alpha$ in iFNO (Freq) across various tasks.

| Tasks | Selection of $\alpha$ | | | | | |
|---|---|---|---|---|---|---|
| | 0.6 | 0.8 | 0.9 | 0.99 | 0.999 | 0.9999 |
| **Burgers** (1e-3) | 1.723 | 0.975 | 0.583 | 0.544 | 0.559 | 0.587 |
| **Darcy Flow** (1e-4) | 0.621 | 0.595 | 0.552 | 0.509 | 0.475 | 0.513 |
| **Navier Stokes (FNO 2D)** | 0.341 | 0.256 | 0.223 | 0.189 | 0.194 | 2.010 |
| **Navier Stokes (FNO 3D)** | 0.524 | 0.499 | 0.461 | 0.464 | 0.468 | 0.471 |
| **Kolmogorov Flow** | 0.133 | 0.122 | 0.102 | 0.043 | 0.049 | 0.051 |

Table 5: Evaluation of $\epsilon$ in iFNO (Loss) across various tasks.

| Tasks | Selection of $\epsilon$ | | | | |
|---|---|---|---|---|---|
| | 0.1 | 0.01 | 0.001 | 0.0001 | 0.00001 |
| **Burgers** (1e-3) | 0.672 | 0.572 | 0.595 | 1.190 | 13.90 |
| **Darcy Flow** (1e-4) | 0.486 | 0.465 | 0.412 | 0.424 | 0.490 |
| **Navier Stokes (FNO 2D)** | 0.192 | 0.194 | 0.191 | 0.284 | 0.322 |
| **Navier Stokes (FNO 3D)** | 0.471 | 0.469 | 0.454 | 0.449 | 0.452 |
| **Kolmogorov Flow** | 0.0614 | 0.0616 | 0.053 | 0.578 | 0.579 |

## B.2 FNO over different sizes

Table 6: Evaluation of changing the width of FNO models on Burger. Validation losses have been divided by 1e-3.

| Models | Dimension | | | | |
|---|---|---|---|---|---|
| | 32 | 64 | 80 | 96 | 128 |
| **iFNO (Freq)** | 0.785 | 0.5901 | 0.532 | **0.469** | **0.513** |
| **iFNO (Loss)** | 0.746 | 0.708 | 0.605 | 0.578 | 0.714 |
| **FNO** | **0.603** | **0.519** | **0.525** | 0.501 | 0.553 |

Table 7: Evaluation of changing the width of FNO models on Darcy. Validation losses have been divided by 1e-4.

| Models | Dimension | | | | |
|---|---|---|---|---|---|
| | 32 | 64 | 80 | 96 | 112 |
| **iFNO (Freq)** | 0.462 | 0.462 | 0.4395 | 0.4304 | 0.4479 |
| **iFNO (Loss)** | **0.4549** | **0.4553** | 0.4599 | **0.3859** | **0.4321** |
| **FNO** | 0.4683 | 0.5051 | **0.4116** | 0.4922 | 0.4834 |

Table 8: Evaluation of changing the width of FNO models on Navier Stokes 2D. Validation losses have been divided by 1.

| Models | Dimension | | | | |
|---|---|---|---|---|---|
| | 16 | 20 | 32 | 64 | 80 |
| **iFNO (Freq)** | 0.205 | 0.192 | **0.180** | **0.170** | **0.176** |
| **iFNO (Loss)** | **0.199** | **0.191** | 0.181 | 0.177 | 0.177 |
| **FNO** | 0.203 | 0.198 | 0.182 | 0.182 | 0.177 |

Table 9: Evaluation of changing the width of FNO models on Kolmogorov Flow. Validation losses have been divided by 1e-1.

| Tasks | Dimension | | | | |
|---|---|---|---|---|---|
| | 16 | 20 | 32 | 64 | 80 |
| **iFNO (Freq)** | 1.513 | 1.450 | 1.312 | 0.634 | (NR) |
| **iFNO (Loss)** | **0.879** | **0.821** | **0.654** | **0.534** | (NR) |
| **FNO** | 1.072 | 1.024 | 0.872 | 0.743 | 0.721 |

### B.3  Incremental FNO under different initialization scales

Table 10: Evaluation of different initialization scales of FNO/iFNO models on Kolmogorov Flow. We again do a full sweep and report the best values.

| Tasks | Scale of Initialization | | | | |
|---|---|---|---|---|---|
| | 1e-2 | 1e-1 | 1 | 1e1 | 1e2 |
| **iFNO (Freq)** | 0.064 | 0.062 | 0.061 | 0.064 | 0.067 |
| **iFNO (Loss)** | **0.050** | **0.054** | **0.070** | **0.053** | **0.055** |
| **FNO** | 0.073 | 0.073 | 0.072 | 0.072 | 0.073 |

### B.4  Standard FNO with different frequency modes

We conduct more experiments to showcase the classic bias-variance tradeoff in FNOs when increasing the modes.

Table 11: Evaluation of different frequency modes on standard FNO on Kolmogorov Flow. In this task, the minimum value is bolded across each row. Validation losses have been divided by the value next to the dataset across each row.

| Tasks | Frequency Modes | | | | | | | | |
|---|---|---|---|---|---|---|---|---|---|
| | 10 | 20 | 30 | 40 | 60 | 80 | 90 | 100 | 120 |
| **Kolmogorov Flow (1e-1)** | 1.327 | 0.906 | 0.674 | 0.598 | 0.482 | 0.509 | **0.451** | 0.471 | 0.485 |

### B.5  Number of Paramaters

We have compiled a summary of our results comparing iFNO to FNO (90 modes) across various tasks:

1. **Burgers equation**:
   - **Generalization**: iFNO improves testing error by 11

- **Efficiency**: iFNO is 21
- **Model** size: iFNO achieves a 1.3x reduction (7.9 million vs. 12 million parameters).

2. **Darcy Flow**:

   - **Generalization**: iFNO improves testing error by 31
   - **Efficiency**: iFNO is 44
   - **Model size**: iFNO achieves a 2.3x reduction (471 million vs. 1.061 billion parameters).

3. **Navier-Stokes 2D+time**:

   - **Generalization**: iFNO (Resolution) improves testing error by 38
   - **Efficiency**: iFNO is 38
   - **Model size**: iFNO achieves a 2.3x reduction (471 million vs. 1.061 billion parameters).

4. **Navier-Stokes 3D**:

   - **Generalization**: iFNO (freq) improves testing error by 31
   - **Efficiency**: iFNO is 31
   - **Model size**: iFNO achieves a 2.8x reduction (1.25 billion vs. 3.539 billion parameters).

5. **Navier-Stokes High Resolution**:

   - **Generalization**: iFNO (Resolution) improves testing error by 3
   - **Efficiency**: iFNO is 46
   - **Model size**: iFNO achieves a 2.3x reduction (471 million vs. 1.061 billion parameters).

These results demonstrate that iFNO consistently outperforms FNO (90 modes) in terms of generalization, efficiency, and model size across various tasks. We chose to compare primarily with FNO (90 modes) as it represents the most challenging baseline. Other iFNO variants, while potentially larger than the base iFNO model, still maintain efficiency advantages over FNO.

Table 12: Evaluation and number of parameters (in Millions) for the various models in Table 3.

| Method | Navier Stokes (FNO 2D) | | Kolmogorov Flow | |
| --- | --- | --- | --- | --- |
| | Number of Parameters (Millions) | Test ($1e-1$) | Number of Parameters (Millions) | Test ($1e-2$) |
| iFNO (Freq) | 0.92M | **1.872** | 321M | **4.602** |
| iFNO (Loss) | 23.8M | 1.911 | 530M | 5.127 |
| FNO (10) | **0.64M** | 1.941 | **6.57M** | 13.27 |
| FNO (30) | 5.76M | 1.924 | 59.0M | 6.462 |
| FNO (60) | 23.0M | 1.918 | 235M | 4.825 |
| FNO (90) | 51.8M | 1.918 | 530M | 4.596 |

## B.6  Tensorized Factorized FNO under different tensor ranks

iTFNO incorporates our incremental mode selection and incremental resolution techniques into the TFNO framework. Specifically, we apply the following modifications to TFNOs:

1. **Incremental mode selection:** Similar to iFNO, iTFNO starts with a small number of modes and gradually increases the number of modes during training. We experiment with both frequency-based (iFNO (Freq)) and loss-based (iFNO (Loss)) criteria for determining when to add modes. This allows iTFNO to adaptively adjust the model capacity and focus on the most relevant frequency components for the given task.

2. **Incremental resolution:** iTFNO also adopts the incremental resolution technique from iFNO, where the model is trained on progressively higher resolutions of the input data. This enables iTFNO to capture fine-grained details and scale to higher resolutions more efficiently.

3. **Factorization schemes and ranks:** In addition to the incremental techniques, we explore different factorization schemes and ranks for the tensor decomposition of the weights in iTFNO. This allows us to investigate the impact of the tensor structure on the performance and efficiency of the incremental approach.

Table 13: Evaluation of 2 different factorization ranks of iTFNO models on Re5000. We do a full sweep and report the best values.

| Weight Factorization | 5% | | 25% | |
|---|---|---|---|---|
| | Test (1e-1) | Runtime (Mins) | Test (1e-1) | Runtime (Mins) |
| **iTFNO** | **0.646** | **122** | **0.793** | **259** |
| **iTFNO (Freq)** | 0.752 | 144 | 0.904 | 333 |
| **TFNO** | 0.719 | 166 | 0.872 | 280 |

## B.7 Different Losses

We have also conducted our models on the $H_1$ Loss. We acknowledge that the $L_2$ loss, while commonly used, may not fully capture the desired properties and behavior of the learned operators. The $L_2$ loss treats all parts of the signal equally and may not prioritize important structural or geometric features. Our proposed iFNO(freq) and iFNO(loss) methods are closely tied to controlling the $L_2$ loss, either directly or indirectly.

We have conducted additional experiments using the Sobolev loss ($H_1$). The Sobolev loss considers not only the pointwise differences between the predicted and target signals but also the differences in their derivatives or gradients. This loss is more sensitive to the smoothness and regularity of the learned operators and can provide insights into their ability to capture important spatial or temporal structures. We have evaluated our proposed methods, including iFNO, iFNO(freq), iFNO(loss), and iFNO(res), as well as the baseline FNO and TFNO models, on three datasets: Re5000, Navier-Stokes 2D, and Burgers. The results are presented in the tables above. For the Re5000 dataset, we observe that iTFNO outperforms both iTFNO(freq) and the baseline TFNO in terms of $H_1$ loss, indicating that the incremental approach combined with the tensor factorization can lead to improved accuracy in capturing the spatial regularity of the solutions. The performance gap is consistent across different tensor ranks (5 and 25).

Table 14: Comparison of iTFNO and TFNO

| | rank 5 | rank 25 |
|---|---|---|
| iTFNO | **0.2396** | **0.2984** |
| iTFNO (Freq) | 0.2889 | 0.3321 |
| TFNO | 0.2725 | 0.3496 |

In the Navier-Stokes 2D dataset, iFNO and its variants (iFNO(freq), iFNO(loss), iFNO(res)) generally achieve lower H1 loss compared to the baseline FNO. This suggests that the incremental mode selection and resolution techniques can help learn operators that better preserve the smoothness and continuity of the fluid dynamics. Similarly, for the Burgers dataset, iFNO and its variants demonstrate competitive or slightly improved performance in terms of $H_1$ loss compared to the baseline FNO. The iFNO(res) variant, which focuses on incremental resolution, achieves the lowest $H_1$ loss, highlighting the importance of adaptively refining the spatial resolution for this specific problem.

Finally, we include one more comparison on the NS2D dataset where we present results for training on $L_2$ loss and evaluating on both $L_2$ and $H_1$ losses, as well as training on $H_1$ loss and evaluating on both $L_2$ and $H_1$

Table 15: NS2D Results

| Model | Test $H_1$ Loss |
|---|---|
| iFNO | 0.1511 |
| iFNO Freq | **0.1476** |
| iFNO Loss | 0.1606 |
| iFNO Res | 0.1507 |
| FNO | 0.1619 |

Table 16: Burgers Results

| Model | Test $H_1$ Loss |
|---|---|
| iFNO | 0.02913 |
| iFNO Freq | 0.02892 |
| iFNO Loss | 0.03005 |
| iFNO Res | **0.02675** |
| FNO | 0.03160 |

losses. The incremental techniques in iFNO and its variants can lead to improved performance in $L_2$ and $H_1$ losses, demonstrating their effectiveness in capturing both pointwise accuracy and spatial regularities. The iFNO variants show different behaviors when trained on $L_2$ versus $H_1$ loss, suggesting that the incremental techniques interact differently with these loss functions.

Table 17: NS2D+Time Results

| Model | Train ($L_2$) | Eval ($L_2$) | Eval ($H_1$) | Train ($H_1$) | Eval ($L_2$) | Eval ($H_1$) |
|---|---|---|---|---|---|---|
| iFNO | 0.0524 | 0.0377 | 0.1284 | 0.0972 | **0.0831** | 0.1543 |
| iFNO Freq | 0.0514 | 0.0369 | 0.1259 | **0.0963** | 0.1244 | **0.1476** |
| iFNO Loss | **0.0504** | **0.0359** | **0.1242** | 0.1144 | 0.1181 | 0.1606 |
| iFNO Res | 0.0546 | 0.0379 | 0.1316 | 0.1152 | 0.0881 | 0.1507 |
| FNO | 0.0517 | 0.0360 | 0.1258 | 0.1103 | 0.1584 | 0.1619 |

These results provide reassurance that our proposed incremental techniques not only improve the $L_2$ loss but also lead to better performance in terms of the Sobolev loss, which is more sensitive to the spatial or temporal regularity of the learned operators. However, we acknowledge that the Sobolev loss is just one example of an alternative metric, and some many other relevant losses and distances could be considered, such as $L_1$ loss, Wasserstein distance, or problem-specific metrics. We believe that a comprehensive evaluation across multiple losses and datasets is crucial for understanding the strengths and limitations of operator learning methods. We will investigate this in future work.

