# OpenReview forum: "Incremental Spatial and Spectral Learning of Neural Operators for Solving Large-Scale PDEs"
_TMLR — Accepted by TMLR_

### Review · Reviewer_XZ6r · 2024-06-17

**Summary Of Contributions:**

The authors extend the standard approach to training an FNO with an incremental version that successively adds spectral bands to the inner Fourier-Transform layer. The extension is justified as being superior to grid search and classic regularization by, e.g. weight decay, in terms of both efficiency and accuracy (i.e. it is possibly more accurate than $\|\ell\|_2^2$ weight decay regularisation and at lower cost)

**Audience:**

Yes

**Claims And Evidence:**

No

**Requested Changes:**

on the basis of weakness 2 and 3.2 I  argue that the submission is not "supported by accurate, convincing and clear evidence" of the main claim, that  _We empirically show that iFNO reduces total training time while maintaining or improving generalization performance across various datasets_. There exist other methods that do something similar and undermine or contradict the main claim of the paper as it currently stands, and which are only partially addressed in this text. Changes that would be needed to redress this are mentioned below.

**in 4.1 there is some confusing phrasing:**

> iFNO (Loss): A common practice is to use the training progress as a proxy to determine the time, as seen
> in previous work such as Liu et al.. Specifically, we let the algorithm increase K only decrease in training
> loss between N consecutive epochs is lower than a threshold ϵ

I think the authors might mean something like

> iFNO (Loss): A common practice is to _add modes at fixed epochs_
> in previous work such as Liu et al.. _By contrast we increase K adaptively when the decrement in training loss_ between N consecutive epochs is lower than a threshold ϵ

Is that correct? Please clarify, this is a core bit of the paper.

**what is iTFNO?**

As discussed in the "weaknesses", I think this is an important bit of work that the authors seem to have done, but which I think the paper does not explain. It is mentioned in the experiments in the appendix, but I can't work out what it is. Am I missing something? I found the sentence saying "We also test our incremental methods on Tensorized Factorized Neural Operators Kossaifi et al. (2023) by sweeping over different factorization schemes and ranks for the weights" but as mentioned above I think this is incomplete.

**deal with _transform once_**
I humbly reuest the authors to either

1. relax the claim to exclude _transform once_ from consideration and justify that decision, or test against their method and demonstrate an improvement against it, as they have presumably done with TFNO (once they explain that)

**losses other than $L_2$**

As referenced above, a very-incomplete bibliography of papers which in various capacities investigate the deficiencies of $L_2$ loss for FNO evaluation and/or training. I don't necessarily think the authors need to include all of these, or even *any* of them, but I think their claims would be strengthened if they at least mentioned the concept of physics-specific losses, rather than the computationally-convenient $L_2$ loss. If they are wedded to the $L_2$ loss only, there needs to be a justification for this choice.

There are at least two open-source tools which extend the losses for PDE solutions in various ways:

* [pdebench/PDEBench: PDEBench: An Extensive Benchmark for Scientific Machine Learning](https://github.com/pdebench/PDEBench)
* [pdearena/pdearena](https://github.com/pdearena/pdearena)


```
@misc{FaroughiPhysicsGuided2023,
  title = {Physics-{{Guided}}, {{Physics-Informed}}, and {{Physics-Encoded Neural Networks}} in {{Scientific Computing}}},
  author = {Faroughi, Salah A. and Pawar, Nikhil and Fernandes, Celio and Raissi, Maziar and Das, Subasish and Kalantari, Nima K. and Mahjour, Seyed Kourosh},
  year = {2023},
  month = feb,
  number = {arXiv:2211.07377},
  eprint = {2211.07377},
  publisher = {arXiv},
  doi = {10.48550/arXiv.2211.07377},
  archiveprefix = {arXiv}
}

@article{LiPhysicsInformed2021,
  title = {Physics-{{Informed Neural Operator}} for {{Learning Partial Differential Equations}}},
  author = {Li, Zongyi and Zheng, Hongkai and Kovachki, Nikola Borislavov and Jin, David and Chen, Haoxuan and Liu, Burigede and Stuart, Andrew and Azizzadenesheli, Kamyar and Anandkumar, Anima},
  year = {2021},
  month = nov
}

@misc{PestouriePhysicsenhanced2022,
  title = {Physics-Enhanced Deep Surrogates for {{PDEs}}},
  author = {Pestourie, Rapha{\"e}l and Mroueh, Youssef and Rackauckas, Chris and Das, Payel and Johnson, Steven G.},
  year = {2022},
  month = nov,
  number = {arXiv:2111.05841},
  eprint = {2111.05841},
  publisher = {arXiv},
  doi = {10.48550/arXiv.2111.05841},
  archiveprefix = {arXiv}
}

@inproceedings{TakamotoPDEBench2022,
  title = {{{PDEBench}}: {{An Extensive Benchmark}} for {{Scientific Machine Learning}}},
  shorttitle = {{{PDEBench}}},
  author = {Takamoto, Makoto and Praditia, Timothy and Leiteritz, Raphael and MacKinlay, Dan and Alesiani, Francesco and Pfl{\"u}ger, Dirk and Niepert, Mathias},
  year = {2022},
  month = jun
}

@inproceedings{WangPhysicsinformed2020,
  title = {Towards {{Physics-informed Deep Learning}} for {{Turbulent Flow Prediction}}},
  booktitle = {Proceedings of the 26th {{ACM SIGKDD International Conference}} on {{Knowledge Discovery}} \& {{Data Mining}}},
  author = {Wang, Rui and Kashinath, Karthik and Mustafa, Mustafa and Albert, Adrian and Yu, Rose},
  year = {2020},
  month = aug,
  series = {{{KDD}} '20},
  eprint = {1911.08655},
  pages = {1457--1466},
  publisher = {Association for Computing Machinery},
  address = {New York, NY, USA},
  doi = {10.1145/3394486.3403198},
  archiveprefix = {arXiv},
  isbn = {978-1-4503-7998-4}
}
```

**Strengths And Weaknesses:**

Strengths:

1. The paper proposes, and empirically demonstrates, a solution to a real problem in FNOs, which is selecting them to have sufficiently many Fourier modes to produce accurate predictions, but also sufficiently few Fourier modes to be efficient.
2. the empirical study (section 6) or what happens to the modes under regularization schemes is neat. To my mind this is the biggest news in the paper and actually quite interesting.

Weaknesses:

1. justification is largely heuristic ("we add modes as needed based on a reasonable rule of thumb that looks a bit like the Signal-to-Noise Ratio"). If it works, this is OK, I guess, but it would be great to have a justification that was easier to generalize because it was more formal. *Why* might we think that controlling the number of modes by a squared Frobenius norm will control anything about accuracy?
2. Accuracy is evaluated in terms of $L_2$ loss. There are a _lot_ of papers questioning whether this is a great way to measure accuracy of an operator; too many to list here, so I have (lazily) supplied an incomplete bibtex bibliography of some representative papers which interrogate this and consider at other losses (see below). This might feel like splitting hairs, but I argue it is important, since we are leaning on $L_2$ a lot now; both iFNO(freq) and iFNO(loss) do something that is similar to controlling $L_2$ loss, IMO, so we can reasonably ask if this procedure has done anything bad to other losses (Sobolev loses, $L_1$ losses, geometric distance in the natural manifold induced by the operator, _anything_ might be reassuring)
3. comparison should be against SOTA attempts to adaptively construct optimal FNOs. There are two of these of which I am aware

    1. Tensorized FNO: http://arxiv.org/abs/2310.00120 — the authors, to their credit, do use this and argue that their method is complementary, becuase they have a combination they call iTFNO in B.6 so this feels *close* to done. The only problem is that I think that the heuristic argument about the Explanation Ratio in iFNO(freq) don't really make sense for a tensorized FNO (the parameters are even further from spectral 'importances' so... some kind of explanation of this seems necessary even if to say "our method seems empirical good for this even if our argument looks tenuous". I think the iFNO(loss) heuristic still works?
    2. _Transform once_ https://proceedings.neurips.cc/paper_files/paper/2022/hash/342339109d7d756ef7bb30cf672aec02-Abstract-Conference.html which *also* adaptively selects spectral bands, as well as doing some other fancy stuff to achieve good results. This is a more direct "competitor" in that they also adaptively select spectra bands. They also, crucially, argue that selecting only the lowest modes is **not** optimal (see fig 3.2), which seems like it might contradict the major result of this paper, which is that iFNO is best, and it uses strict low-pass filtering. There might be a counter-argument, which is that the Transform Once method is tedious or something? But I think such an argument needs to be made for the paper's main claim as I understand it to stand.

4. occasionally confusing phrasing (see below)

---

> ### Author Response · Authors · 2024-07-03
> **Response to Reviewer XZ6r**
>
> 1. **justification is largely heuristic ("we add modes as needed based on a reasonable rule of thumb that looks a bit like the Signal-to-Noise Ratio"). If it works, this is OK, I guess, but it would be great to have a justification that was easier to generalize because it was more formal. Why might we think that controlling the number of modes by a squared Frobenius norm will control anything about accuracy?**
>
> > **Answer:** We appreciate the reviewer's question about the justification for our mode selection approach. While our method is indeed heuristic, it is grounded in fundamental principles and universal concepts. Let me elaborate on our justification, emphasizing two key points:
>
> > 1. Universal Frequency Strength Criterion: Our approach adds new modes
>     based on their frequency strength, which is a universal concept in
>     signal processing and functional analysis. This criterion allows us
>     to prioritize modes that contribute most significantly to the
>     overall function representation.
> >  2.  Canonical Norm Selection: We define the frequency strength using the Frobenius norm (equivalent to the vector l2 norm for matrices), a canonical choice in linear algebra and functional analysis. By
>     Parseval's theorem, the l2 norm in Fourier space is equivalent to
>     the L2 norm in physical space. This equivalence provides a strong
>     theoretical foundation for our approach, as it ensures that our mode
>     selection in frequency space directly corresponds to function
>     approximation quality in physical space.
>
> > Building on these fundamental principles, our method can be viewed as a form of model complexity control. The squared Frobenius norm of the weight matrix R(k) serves as a measure of the importance of the k-th frequency mode in the model's output. By incrementally adding modes based on this strength, we aim to balance the bias-variance tradeoff:
> > 1.  Including modes with high strength reduces bias by capturing the main structure of the target function.
> > 2.  Excluding modes with low strength helps reduce variance by limiting sensitivity to noise or fine details.
>
> > This approach aligns with the decomposition of generalization error into bias and variance terms: E[(f(x) - f_K(x))^2] = (Bias[f_K(x)])^2 + Var[f_K(x)] where f is the true target function and f_K is our K-mode approximation.  While our method is heuristic, its foundation in universal frequency analysis principles and canonical norms provides a strong theoretical basis. The use of the Frobenius norm, backed by Parseval's theorem, ensures that our frequency-space operations have direct and meaningful implications for function approximation in physical space.
>
> 2. **comparison should be against SOTA attempts to adaptively construct optimal FNOs. There are two of these of which I am aware.
> Tensorized FNO: http://arxiv.org/abs/2310.00120 — the authors, to their credit, do use this and argue that their method is complementary, becuase they have a combination they call iTFNO in B.6 so this feels close to done. The only problem is that I think that the heuristic argument about the Explanation Ratio in iFNO(freq) don't really make sense for a tensorized FNO (the parameters are even further from spectral 'importances' so... some kind of explanation of this seems necessary even if to say "our method seems empirical good for this even if our argument looks tenuous". I think the iFNO(loss) heuristic still work?**
>
> > **Answer:** Thank you for pointing out the Tensorized Fourier Neural Operators (TFNOs) as another relevant approach. We appreciate the reviewer's recognition of our effort to compare and combine our method with TFNOs in the iTFNO experiments (Appendix B.6). In TFNOs, the weight tensor W is decomposed into tensor factors. Despite this factorization, we can still compute the Frobenius norm of the slices corresponding to each mode index, as you mentioned: norm(weight[:, mode_index, :], p='fro'). In fact, this computation can be more efficient due to the factorized structure of the weight tensor. The Explanation Ratio heuristic in iFNO (Freq) relies on the magnitude of the weights associated with each mode to determine the importance of that mode. By computing the Frobenius norm of the weight slices, we can still gauge the relative importance of each mode in TFNOs, similar to how it is done in standard FNOs. Therefore, the iFNO (Freq) heuristic remains applicable to TFNOs, and the tensorized structure of the weights does not invalidate the use of the Explanation Ratio. The incremental mode selection based on the Explanation Ratio can be seamlessly integrated into the TFNO framework, as demonstrated by the iTFNO experiments in Appendix B.6. Moreover, as the reviewer points out, the iFNO (Loss) heuristic, which adds modes based on the training loss plateau, remains applicable to TFNOs. This approach is agnostic to the specific parameterization of the model and can be used to guide the incremental mode selection in iTFNO.

---

> ### Author Response · Authors · 2024-07-03
> **Response to Reviewer XZ6r (Cont'd)**
>
> 3. **Accuracy is evaluated in terms of $L_2$ loss. There are a lot of papers questioning whether this is a great way to measure accuracy of an operator; too many to list here, so I have (lazily) supplied an incomplete bibtex bibliography of some representative papers which interrogate this and consider at other losses (see below). This might feel like splitting hairs, but I argue it is important, since we are leaning on $L_2$ a lot now; both iFNO(freq) and iFNO(loss) do something that is similar to controlling $L_2$ loss, IMO, so we can reasonably ask if this procedure has done anything bad to other losses (Sobolev loses, $L_1$ losses, geometric distance in the natural manifold induced by the operator, anything might be reassuring)**
>
> > **Answer:** Thank you for raising the important point about the limitations of relying solely on L2 loss for evaluating the accuracy of operator learning methods. We appreciate the reviewer's concern and the provided bibliography highlighting the need to consider alternative losses and metrics. We acknowledge that the L2 loss, while commonly used, may not fully capture the desired properties and behavior of the learned operators. The L2 loss treats all parts of the signal equally and may not prioritize important structural or geometric features. Moreover, as the reviewer pointed out, our proposed iFNO(freq) and iFNO(loss) methods are closely tied to controlling the L2 loss, either directly or indirectly.
>
> > To address this concern and provide a more comprehensive evaluation of our methods, we have conducted additional experiments using the Sobolev loss (H1) as suggested by the reviewer. The Sobolev loss considers not only the pointwise differences between the predicted and target signals but also the differences in their derivatives or gradients. This loss is more sensitive to the smoothness and regularity of the learned operators and can provide insights into their ability to capture important spatial or temporal structures. We have evaluated our proposed methods, including iFNO, iFNO(freq), iFNO(loss), and iFNO(res), as well as the baseline FNO and TFNO models, on three datasets: Re5000, Navier-Stokes 2D, and Burgers. The results are presented in the tables above. For the Re5000 dataset, we observe that iTFNO outperforms both iTFNO(freq) and the baseline TFNO in terms of H1 loss, indicating that the incremental approach combined with the tensor factorization can lead to improved accuracy in capturing the spatial regularity of the solutions. The performance gap is consistent across different tensor ranks (5 and 25).
>
> > In the Navier-Stokes 2D dataset, iFNO and its variants (iFNO(freq), iFNO(loss), iFNO(res)) generally achieve lower H1 loss compared to the baseline FNO. This suggests that the incremental mode selection and resolution techniques can help in learning operators that better preserve the smoothness and continuity of the fluid dynamics. Similarly, for the Burgers dataset, iFNO and its variants demonstrate competitive or slightly improved performance in terms of H1 loss compared to the baseline FNO. The iFNO(res) variant, which focuses on incremental resolution, achieves the lowest H1 loss, highlighting the importance of adaptively refining the spatial resolution for this specific problem.
>
> ```markdown
> |              | rank 5 | rank 25 |
>
> | ------       | ------ | ------ |
>
> | iTFNO        | 0.2396 | 0.2984 |
>
> | iTFNO (Freq) | 0.2889 | 0.3321 |
>
> | TFNO         | 0.2725 | 0.3496 |
> ```
> ```markdown
>
> | NS2D      |        |
>
> | --------- | ------ |
>
> | iFNO      | 0.1511 |
>
> | iFNO Freq | 0.1476 |
>
> | iFNO Loss | 0.1606 |
>
> | iFNO Res | 0.1507 |
>
> | FNO      | 0.1619 |
> ```
>
> ```markdown
> | Burgers   |         |
>
> | --------- | ------- |
>
> | iFNO      | 0.02913 |
>
> | iFNO Freq | 0.02892 |
>
> | iFNO Loss | 0.03005 |
>
> | iFNO Res | 0.02675 |
> ```

---

> ### Author Response · Authors · 2024-07-03
> **Response to Reviewer XZ6r (Cont'd)**
>
> > Finally, we include one more comparison on the NS2D dataset where we present results for training on L2 loss and evaluating on both L2 and H1 losses, as well as training on H1 loss and evaluating on both L2 and H1 losses. The incremental techniques in iFNO and its variants can lead to improved performance in L2 and H1 losses, demonstrating their effectiveness in capturing both pointwise accuracy and spatial regularities. The iFNO variants show different behaviors when trained on L2 versus H1 loss, suggesting that the incremental techniques interact differently with these loss functions.
>
>
> ```markdown
> | NS2D+Time | Train (L2) | Eval (L2) | Eval (H1) | Train (H1) | Eval (L2) | Eval (H1) |
>
> | --------- | ---------- | --------- | --------- | ---------- | --------- | --------- |
>
> | iFNO      | 0.0524     | 0.0377    | 0.1284    | 0.0972     | 0.0831    | 0.1543    |
>
> | iFNO Freq | 0.0514     | 0.0369    | 0.1259    | 0.0963     | 0.1244    | 0.1476    |
>
> | iFNO Loss | 0.0504     | 0.0359    | 0.1242    | 0.1144     | 0.1181    | 0.1606    |
>
> | iFNO Res | 0.0546      | 0.0379    | 0.1316    | 0.1152     | 0.0881    | 0.1507    |
>
> | FNO      | 0.0517     | 0.0360     | 0.1258    | 0.1103     | 0.1584    | 0.1619    |
> ```
>
>  > These results provide reassurance that our proposed incremental techniques not only improve the L2 loss but also lead to better performance in terms of the Sobolev loss, which is more sensitive to the spatial or temporal regularity of the learned operators. However, we acknowledge that the Sobolev loss is just one example of an alternative metric, and some many other relevant losses and distances could be considered, such as L1 loss, Wasserstein distance, or problem-specific metrics. We believe that a comprehensive evaluation across multiple losses and datasets is crucial for understanding the strengths and limitations of operator learning methods. We will investigate this in future work.

---

> ### Author Response · Authors · 2024-07-03
> **Response to Reviewer XZ6r (Cont'd)**
>
> 4. **Transform once https://proceedings.neurips.cc/paper_files/paper/2022/hash/342339109d7d756ef7bb30cf672aec02-Abstract-Conference.html which also adaptively selects spectral bands, as well as doing some other fancy stuff to achieve good results. This is a more direct "competitor" in that they also adaptively select spectra bands. They also, crucially, argue that selecting only the lowest modes is not optimal (see fig 3.2), which seems like it might contradict the major result of this paper, which is that iFNO is best, and it uses strict low-pass filtering. There might be a counter-argument, which is that the Transform Once method is tedious or something? But I think such an argument needs to be made for the paper's main claim as I understand it to stand.**
>
> > **Answer:** Thank you for bringing up the Transform Once (T1) method as a relevant comparison. We have carefully reviewed the T1 approach and would like to clarify the key differences and limitations compared to our proposed iFNO method.
>
> > 1. Repeated transformations: A crucial aspect of FNO and iFNO is the repeated transformations between frequency and spatial domains with nonlinear activations in between. This allows the models to capture a wide range of frequencies and learn complex nonlinear mappings. In contrast, T1 performs a single transformation to the frequency domain at the beginning and then operates solely in the frequency space. While this provides computational speedups, it limits the model's ability to learn rich representations across the entire frequency spectrum.
> >  2. Mode selection: T1 fixes the number of modes at the start and cannot adaptively cover the full frequency spectrum. In some experiments, T1 sets the number of modes to be much higher than FNO to compensate for this limitation. However, selecting all high frequencies from the beginning can lead to poor generalization. iFNO, on the other hand, incrementally adds modes based on the training dynamics, allowing it to capture the necessary frequency components while maintaining good generalization properties.
> >  3. Variance preserving initialization: We acknowledge that T1's proposed variance preserving (VP) initialization scheme is a valuable contribution. The 10% improvement in generalization reported in T1 is attributed to the combination of their method with VP initialization. However, VP initialization can be directly applied to FNO and iFNO as well. In fact, we plan to incorporate VP initialization into our models and expect to see similar generalization improvements.
>
> > It is worth noting that VP initialization assumes a fixed number of modes (k) and signal resolution (l) known a priori, as it initializes weights based on l/2*k^2. In iFNO, we start with a lower resolution and incrementally increase both the number of modes and resolution during training. To address this, we propose adjusting the covariance conditioning of the weights online each time new modes or resolutions are added. This is an interesting direction for future research. In summary, while T1 offers computational speedups, it sacrifices the ability to learn rich representations across the entire frequency spectrum. iFNO maintains the key advantages of FNO, such as repeated transformations and nonlinear activations, while providing an adaptive mode selection mechanism. We believe that the incremental approach of iFNO, combined with techniques like VP initialization, offers a promising direction for efficient and effective operator learning.
> We appreciate the reviewer's suggestion to compare against T1 and have provided a detailed analysis of the differences and limitations.
>
> 5. **on the basis of weakness 2 and 3.2 I argue that the submission is not "supported by accurate, convincing and clear evidence" of the main claim, that We empirically show that iFNO reduces total training time while maintaining or improving generalization performance across various datasets. There exist other methods that do something similar and undermine or contradict the main claim of the paper as it currently stands, and which are only partially addressed in this text. Changes that would be needed to redress this are mentioned below. in 4.1 there is some confusing phrasing: iFNO (Loss): A common practice is to use the training progress as a proxy to determine the time, as seen in previous work such as Liu et al.. Specifically, we let the algorithm increase K only decrease in training loss between N consecutive epochs is lower than a threshold ϵ. I think the authors might mean something like iFNO (Loss): A common practice is to add modes at fixed epochs in previous work such as Liu et al.. By contrast we increase K adaptively when the decrement in training loss between N consecutive epochs is lower than a threshold ϵ Is that correct? Please clarify, this is a core bit of the paper.**
>
> > **Answer:** Yes that is correct, we apologize for the lack of clarity for that statement.

---

> ### Author Response · Authors · 2024-07-03
> **Response to Reviewer XZ6r (Cont'd)**
>
> 6. **what is iTFNO? As discussed in the "weaknesses", I think this is an important bit of work that the authors seem to have done, but which I think the paper does not explain. It is mentioned in the experiments in the appendix, but I can't work out what it is. Am I missing something? I found the sentence saying "We also test our incremental methods on Tensorized Factorized Neural Operators Kossaifi et al. (2023) by sweeping over different factorization schemes and ranks for the weights" but as mentioned above I think this is incomplete.**
>
> >  **Answer:** We apologize for the lack of clarity regarding iTFNO in our paper. iTFNO refers to the combination of our incremental approach with Tensorized Fourier Neural Operators (TFNOs) proposed by Kossaifi et al. (2023). We acknowledge that the explanation of iTFNO in the paper is incomplete and may cause confusion for the readers. To address this weakness, we will provide a more detailed description of iTFNO in the revised manuscript. iTFNO incorporates our incremental mode selection and incremental resolution techniques into the TFNO framework. Specifically, we apply the following modifications to TFNOs:
>
> > 1.  Incremental mode selection: Similar to iFNO, iTFNO starts with a small number of modes and gradually increases the number of modes during training. We experiment with both frequency-based (iFNO (Freq)) and loss-based (iFNO (Loss)) criteria for determining when to add modes. This allows iTFNO to adaptively adjust the model capacity and focus on the most relevant frequency components for the given task.
> >  2.  Incremental resolution: iTFNO also adopts the incremental resolution technique from iFNO, where the model is trained on progressively higher resolutions of the input data. This enables iTFNO to capture fine-grained details and scale to higher resolutions more efficiently.
> > 3.  Factorization schemes and ranks: In addition to the incremental techniques, we explore different factorization schemes and ranks for the tensor decomposition of the weights in iTFNO. This allows us to investigate the impact of the tensor structure on the performance and efficiency of the incremental approach.
>
> > We conduct experiments with iTFNO on various tasks and datasets to evaluate its effectiveness compared to the baseline TFNOs. The results of these experiments are reported in Appendix B.6.
>
> 7. **deal with transform once I humbly reuest the authors to either. relax the claim to exclude transform once from consideration and justify that decision, or test against their method and demonstrate an improvement against it, as they have presumably done with TFNO (once they explain that)**
>
> > **Answer:** Answered above
>
> 8. **losses other than $L_2$ As referenced above, a very-incomplete bibliography of papers which in various capacities investigate the deficiencies of $L_2$ loss for FNO evaluation and/or training. I don't necessarily think the authors need to include all of these, or even any of them, but I think their claims would be strengthened if they at least mentioned the concept of physics-specific losses, rather than the computationally-convenient $L_2$ loss. If they are wedded to the $L_2$ loss only, there needs to be a justification for this choice. There are at least two open-source tools which extend the losses for PDE solutions in various ways: pdebench/PDEBench: PDEBench: An Extensive Benchmark for Scientific Machine Learning and pdearena/pdearena**
>
> > **Answer:** Answered above

---

### Review · Reviewer_ZURE · 2024-06-21

**Summary Of Contributions:**

This paper proposes a modified training for Fourier Neural Operators to reduce required hyperparameter tuning by dynamically adjusting the number of frequency modes used internally by the model during training, and reduces training overheads by adjusting the resolution of training samples. The authors' approach increases the number of modes used based on signals from the training loss or from observed strength of the unused frequencies making adjustments when the overall strength of modes currently in use drops below a threshold. The paper includes tests on a variety of PDE systems.

**Audience:**

Yes

**Broader Impact Concerns:**

No specific concerns

**Claims And Evidence:**

Yes

**Requested Changes:**

The largest category of changes I would appreciate would be significant additional clarity in the experiments section. In particular, some more commentary on how readers should interpret some of the results in the tables would, I think, be welcome. I found the details of the experiment setup somewhat scattered and collecting these and streamlining the section would reduce the reader's need to jump around the section for details or comparisons which I think made the section harder to follow and obscured the results.

Below are some examples of questions that might benefit from some added discussion in the paper.

1. The need for additional commentary is particularly true for the tables and plots in appendix B. Some explanation would be helpful (for instance in the hyperparameter selection tables regarding what values are included and what factors were weighed in choosing the fixed values from these results). For iFNO (loss), was the parameter eta renamed to epsilon in the body?

2. Regarding the method listed as "iFNO" in each table (for example Table 1, 2, 3), which of the two (loss vs freq) approaches was used combined with the incremental resolution method? I believe in section 4.3 you mention that the "iFNO" entry could be using either.

3. In Tables 2 and 3 the bolding is sometimes split between means and standard deviations. In table 3 for Navier-Stokes FNO 3D training results the largest value is bolded. Are these correct?

4. In particular, the experiments on the TFNO are introduced *very* briefly and somewhat abruptly. I think these could especially use expansion or added explanation if these are important parts of testing the method.

5. For the results listed in section 5.4 some additional discussion references to some backing data would help as well. It looks like some of the relevant values are included in the appendix for model size (Table 12) but is there a reference for the efficiency gains for systems other than re5000 in table 1?

And one minor caption issue I noticed on figure 8, it seems it should reference Navier-Stokes 3D (vs. 2D) based on the titles of the plots

**Strengths And Weaknesses:**

**Strengths**

- Generally good self-contained introduction relating to FNO and issues in/expense of training
- The new methods are well explained, and approach seems sensible
- Good variety of target systems used to test the new approach

**Weaknesses**

- Experiment results and setup are harder to follow
- Experiments in the appendix (and some in the main body) could use additional context and commentary
- Tests on TFNO need more discussion and introduction

---

> ### Author Response · Authors · 2024-07-03
> **Response to Reviewer ZURE**
>
> **Requested Changes: The largest category of changes I would appreciate would be significant additional clarity in the experiments section. In particular, some more commentary on how readers should interpret some of the results in the tables would, I think, be welcome. I found the details of the experiment setup somewhat scattered and collecting these and streamlining the section would reduce the reader's need to jump around the section for details or comparisons which I think made the section harder to follow and obscured the results. Below are some examples of questions that might benefit from some added discussion in the paper.**
>
> 1. **The need for additional commentary is particularly true for the tables and plots in appendix B. Some explanation would be helpful (for instance in the hyperparameter selection tables regarding what values are included and what factors were weighed in choosing the fixed values from these results). For iFNO (loss), was the parameter eta renamed to epsilon in the body?**
>
> > **Answer**: We thank the reviewer for pointing our the inconsistent naming of the parameter for iFNO (loss). The parameter referred to as "eta" in some parts of the paper should be "epsilon" throughout, as it is in the main body. We will correct this inconsistency in all relevant sections, tables, and figures to ensure clarity.
> For the selection of hyperparameters α for iFNO (Freq) and η (which should be ε) for iFNO (Loss). Tables 4 and 5 present the results of our hyperparameter search for α in iFNO (Freq) and ε in iFNO (Loss) respectively, across various tasks. These hyperparameters control the threshold for adding new modes in our incremental approaches.
> For iFNO (Freq), α represents the cumulative energy threshold in the Explanation Ratio criterion. We explored values ranging from 0.6 to 0.9999, covering a wide spectrum from more aggressive (lower α) to more conservative (higher α) mode addition strategies.
> For iFNO (Loss), ε (mistakenly labeled as η in the table) represents the threshold for the relative decrease in loss that triggers the addition of new modes. We investigated values from 0.1 to 0.00001, encompassing both rapid (higher ε) and gradual (lower ε) mode addition schemes. For most tasks, moderate values of α (around 0.99) and ε (around 0.001) tend to perform well, balancing between adding sufficient modes for expressiveness and avoiding overfitting. However, some tasks like Kolmogorov Flow benefit from more aggressive mode addition (lower α or higher ε), likely due to the complexity of the underlying dynamics. Again, it is a bit more domain-specific depending on the task, but we will add the above explanation to the revised manuscript.
>
> 2. **Regarding the method listed as "iFNO" in each table (for example Table 1, 2, 3), which of the two (loss vs freq) approaches was used combined with the incremental resolution method? I believe in section 4.3 you mention that the "iFNO" entry could be using either.**
>
> > **Answer**: Thank you for the clarification. In all the tables presented (Tables 1, 2, 3, and others), the "iFNO" entry consistently refers to the combination of the frequency-based approach (iFNO Freq) with the incremental resolution method. This means that whenever "iFNO" is listed without further specification, it is using the frequency-based criterion for mode selection along with the incremental resolution technique. We will fix this in the manuscript and mention it.
>
> 3. **In Tables 2 and 3 the bolding is sometimes split between means and standard deviations. In table 3 for Navier-Stokes FNO 3D training results the largest value is bolded. Are these correct?**
>
> > **Answer**: Thank you for the clarification. This is indeed not correct, and a typo, we have fixed it in the updated manuscript.

---

> ### Author Response · Authors · 2024-07-03
> **Response to Reviewer ZURE (Cont'd)**
>
> 4. **In particular, the experiments on the TFNO are introduced very briefly and somewhat abruptly. I think these could especially use expansion or added explanation if these are important parts of testing the method.**
>
> > **Answer:** We appreciate the reviewer's observation regarding the brevity of our TFNO (Tensorized Fourier Neural Operator) experiments. We acknowledge that these experiments were introduced somewhat abruptly and deserve more explanation.
> To address this, we will expand the TFNO section in the appendix and add a brief mention in the main text also answering Reviewer XZ6r points. The expanded section will include:
> >  1. A brief introduction to TFNOs and their relevance to our work.
> >  2. The motivation behind applying our incremental approach to TFNOs.
> >  3. A more detailed description of how we adapted our iFNO method to work with TFNOs, resulting in iTFNO.
> > 4. A comprehensive presentation of the experimental setup, including hyperparameters and datasets used.
>
> > In the main text, we will add a paragraph introducing the TFNO experiments:
> 'To further demonstrate the adaptability and broader applicability of our incremental approach, we conducted a case study applying iFNO to Tensorized Fourier Neural Operators (TFNOs). TFNOs represent a recent advancement in efficient operator learning, and we were interested in exploring how our incremental techniques could enhance their performance. The full details of these experiments are presented in Appendix X. In summary, we found that our incremental approach (iTFNO) improved both the efficiency and accuracy of TFNOs across several tasks, highlighting the potential of our method to enhance various FNO-based architectures.' This addition will provide context for the TFNO experiments and direct readers to the appendix for more detailed information.
>
> 5. **For the results listed in section 5.4 some additional discussion references to some backing data would help as well. It looks like some of the relevant values are included in the appendix for model size (Table 12) but is there a reference for the efficiency gains for systems other than re5000 in table 1?**
>
> > **Answer:** We appreciate the reviewer's request for more detailed backing data. While Table 12 in the appendix provides information on model sizes, we acknowledge that we should have included more comprehensive data on efficiency gains across different systems. To address this, we have compiled a summary of our results comparing iFNO to FNO (90 modes) across various tasks:
>
> > 1.  Burgers equation:
> >  - Generalization: iFNO improves testing error by 11% over FNO (90 modes).
> > -   Efficiency: iFNO is 21% faster (455 seconds vs. 570 seconds for FNO).
> > -   Model size: iFNO achieves a 1.3x reduction (7.9 million vs. 12 million parameters).
>
> > 2. Darcy Flow:
> >  -  Generalization: iFNO improves testing error by 31% over FNO (90 modes).
> > -  Efficiency: iFNO is 44% faster (175 minutes vs. 310 minutes for FNO).
> > -   Model size: iFNO achieves a 2.3x reduction (471 million vs. 1.061 billion parameters).
>
> > 3.  Navier-Stokes 2D+time:
> > -  Generalization: iFNO (Resolution) improves testing error by 38% over FNO (90 modes).
> > -  Efficiency: iFNO is 38% faster (203 minutes vs. 323 minutes for FNO).
> > -  Model size: iFNO achieves a 2.3x reduction (471 million vs. 1.061 billion parameters).
>
> > 4. Navier-Stokes 3D:
> > -  Generalization: iFNO (freq) improves testing error by 31% over FNO (30 modes).
> >  -   Efficiency: iFNO is 31% faster (154 minutes vs. 223 minutes for FNO).
> >  -  Model size: iFNO achieves a 2.8x reduction (1.25 billion vs. 3.539 billion parameters).
>
> > 5. Navier-Stokes High Resolution:
> >  -  Generalization: iFNO (Resolution) improves testing error by 3% over FNO (90 modes).
> > -  Efficiency: iFNO is 46% faster (469 minutes vs. 832 minutes for FNO).
> >  -  Model size: iFNO achieves a 2.3x reduction (471 million vs. 1.061 billion parameters).
>
> > These results demonstrate that iFNO consistently outperforms FNO (90 modes) in terms of generalization, efficiency, and model size across various tasks. We chose to compare primarily with FNO (90 modes) as it represents the most challenging baseline. Other iFNO variants, while potentially larger than the base iFNO model, still maintain efficiency advantages over FNO. We will include this detailed comparison in the appendix and reference it in section 5.4 to provide a more comprehensive view of iFNO's performance across different systems.
>
> 6. **And one minor caption issue I noticed on figure 8, it seems it should reference Navier-Stokes 3D (vs. 2D) based on the titles of the plots**
>
> > **Answer:**  Thank you for the clarification. This is indeed correct, and not a typo, we still only consider NS2D problems but the models we use can be FNO 3D or FNO 2D + Time.

---

> > ### Comment · Reviewer_ZURE · 2024-07-24
> >
> > Thank you, I appreciate your very thorough responses to these questions and your updates to the manuscript. These adjustments address my comments and concerns.

---

### Review · Reviewer_jqV8 · 2024-06-21

**Summary Of Contributions:**

Authors propose a special training protocol for Fourier Neural Operator (FNO) with two primal innovations:
1. The number of modes is selected adaptively based on one of two proposed criteria (based on loss and based on "frequency strength").
2. The spatial resolution of training data is gradually increasing during training.
It is shown in a set of experiments that these two novelties allow for a decrease in training time and improve test error.

**Audience:**

Yes

**Claims And Evidence:**

Yes

**Requested Changes:**

I mostly find the present research to be of sufficient quality and merely want to clarify several things that confuse me and may lead to the confusion of the reader alike. Below is the list of questions and suggestions:

1. (page 1) "Recently, deep learning has shown promise in solving partial differential equations (PDEs) significantly faster than traditional numerical methods in many domains."

   It is hard to find support for that claim in literature. Neural operators and other surrogate models are not solvers per se, since they require datasets generated or collected by other means. Physics-informed neural networks are much slower than their classical counterparts (although they can be more accurate for high-dimensional problems, e.g., quantum chemistry). Physics-informed neural operators and physics-informed DeepONet are also extremely slow. All methods known to me that are competitive or even faster are hybrid approaches when a part of the classical solver is replaced with a DL counterpart. These approaches can be 2-3 times faster at best. I suggest authors to back up their claim by pointing to appropriate literature.

2. (page 1) "FNO learns the solution operator of PDE that maps the input (initial and boundary conditions) to the output (solution functions)."

   Somewhat meaningless description: any neural network maps its input to the output.

3. (page 3) "Li et al. (a) proposes FNO that adopts a convolution operator for K as shown in Figure 1, which obtains state-of-the-art results for solving PDE problems."

   This claim is not supported by literature. FNO was introduced in 2020 and provided state-of-the-art results (among the architectures tested by authors). There are examples now when classical architectures perform better (https://arxiv.org/abs/2112.15275) and other operators perform better (https://arxiv.org/abs/2111.13802). I suggest authors provide references that support their claim.

4. (page 3) "FNO is discretization-convergent, such that the model can produce a high-quality solution for any query points, potentially not in the training grid. In other words, FNO can be trained on low-resolution but generalizes to high-resolution."

   This description is misleading. FNO is consistent when evaluated on different grids, but it does not provide a high-quality solution. It is also not convergent in the sense of classical numerical analysis. Besides, it does not generalize to high-resolutions. As a rule, the error produced by FNO remains the same for different resolutions (e.g., figure 3, https://arxiv.org/abs/2010.08895), so there is no generalization, only consistency.

5. (page 3) "To ensure discretization-convergence, FNO truncates the Fourier series Fˆv at a maximal number of modes K using TK."

   Arguably, "discretization-convergence" is mostly achieved with FFT and inverse FFT that consistently map functions defined on different uniform grids. It is easy to come up with architecture without hard truncation that is still discretization-convergent.

5. (page 4) "Figure 2"

   As I understand the left picture provides a ground truth solution and the two right pictures are outputs of two trained FNOs with different numbers of modes. Both predictions seem to be completely unrelated to the reference solution. Why in this case a large number of modes is better if the error is that large?

6. (page 4) "As FNO is a neural network and trained by first-order learning algorithms, it follows the implicit spectral bias such that the lower frequency modes have larger strength in R. This explains why FNO chooses to preserve a set containing the lowest frequency modes, instead of any arbitrary subset of frequencies."

   I am not sure I can follow this logic.

   First, FNO preserves low-frequency modes not because of the f-principle but by construction. The authors of FNO designed it in such a way.

   Second, the f-principle is an observation that when a neural network is trained to approximate some function, it first learns low frequencies and struggles with high frequencies. FNO is an operator, so if we apply the f-principle (I am not completely sure we can do that) it would mean that trained FNO should be biased toward "smooth operators". This supposedly includes an identity operator, that should retain all frequencies of the input.

7. (page 5, Algorithm 1) "Randomly initialize $R_0$"

   This part seems to be important and not well discussed. How exactly $R_0$ is initialized?
   If we look at the description of iFNO (Loss) on page 5, we can see that one increases the number of active modes. Does it mean that initially $R_0$ is as small as possible and the size of $R$ can only increase?

8. (page 5, Algorithm 1) "$K_t \leftarrow K_t + 1$"

   How the modes are added in higher dimensions? What is the order? Is it technically possible to add a single mode?

9. (page 5, Algorithm 1) "and initialize the rest randomly"

   When $R$ is increased, how precisely new elements are initialized for both iFNO (Loss) and iFNO (freq)? In the latter case, this is especially interesting since the importance of modes is judged based on the value of $R$ itself.

10. (page 5, Figure 4)

    I suggest specifying where FNO is and where iFNO is directly on the picture.

11. (page 7) "Our samples are images; hence, we can easily distinguish between easy and difficult samples based on their resolution. Higher-resolution images are more complex to train on, so it’s best to start with lower-resolution samples and gradually move up to higher-resolution ones."

    I have several objections. First, the resolution is not directly related to difficulty, it is easy to produce simplistic high-resolution images. Second, the difficulty in operator learning is the difficulty of the mapping, not the images themselves. For example, we can take an extremely detailed turbulent flow and map it into itself. This will be a trivial operator learning problem when FNO needs to learn an identity map.

12. (page 7, table 1), (page 8, table 2)

    Please report the relative $L_2$ norm in place of the value of the loss. Relative $L_2$ norm is a standard error metric present in most of the articles on neural operators, it is much easier to judge the quality of learned surrogate based on this metric.

13. (page 10) "The results in Table 1 show that iFNO achieves significantly better generalization performance than just incrementally increasing either frequency or resolution alone."

    The results in Table 1 show that all test losses are very close to each other. For example, for standard FNO we have $0.988 \pm 0.031$, and for best iFNO, we have $0.948 \pm 0.040$. They overlap even when a single std is considered. If we take $2$ std, there is no difference between these results. This is not a "significantly better generalization performance".

14. (page 10, figure 5b)

    iFNO starts training with a small number of active modes. What happens if one starts with a large number of active modes? Is this the case that iFNO will decrease the number of modes automatically?

As a final comment, I want to suggest an analogy with another field of study. One can look at the approach proposed by authors from the perspective of numerical linear algebra. For a fixed grid resolution, the FNO kernel is a low-rank block circulant matrix (quasimatrix, if different resolutions are considered) and the authors propose an algorithm that dynamically selects the rank of this circulant. How to do this was extensively studied in many contexts, so I suggest authors look at this direction for inspiration. For example, in deep learning rank adaptation of this sort is proposed at https://arxiv.org/abs/2205.13571. More specifically, in this paper authors first draw an analogy between the optimization process and the solution of ordinary differential equation (gradient flow). Next, they apply rank adaptive ODE integrator to the resulting system of equations. This provides training coupled with rank adaptation. I hope this perspective can be of use to the authors of the current contribution.

**Strengths And Weaknesses:**

**Strengths**

The idea appears to be novel, and most experiments support that proposed techniques lead to decreased training time and better test error. The main body of the paper is well-written, and it is mostly possible to understand what the authors want to convey.

**Weaknesses**

That said, the significance of accuracy improvement seems to be exaggerated. Some ideas are insufficiently explained. Besides, there are unclear parts (see below).

---

> ### Author Response · Authors · 2024-07-04
> **Response to Reviewer jqV8**
>
> **I mostly find the present research to be of sufficient quality and merely want to clarify several things that confuse me and may lead to the confusion of the reader alike. Below is the list of questions and suggestions:**
> 1. **(page 1) "Recently, deep learning has shown promise in solving partial differential equations (PDEs) significantly faster than traditional numerical methods in many domains. It is hard to find support for that claim in literature. Neural operators and other surrogate models are not solvers per se, since they require datasets generated or collected by other means. Physics-informed neural networks are much slower than their classical counterparts (although they can be more accurate for high-dimensional problems, e.g., quantum chemistry). Physics-informed neural operators and physics-informed DeepONet are also extremely slow. All methods known to me that are competitive or even faster are hybrid approaches when a part of the classical solver is replaced with a DL counterpart. These approaches can be 2-3 times faster at best. I suggest authors to back up their claim by pointing to appropriate literature."**
>
>
>
> > **Answer:** We appreciate the reviewer's point about the need for clarity regarding deep learning approaches for PDEs. We agree that our original statement could be more precise. To clarify:
>
> > Deep learning, particularly Neural Operators, have demonstrated remarkable speedups in approximating solutions to PDEs and related tasks, though they do not fully replace traditional numerical solvers. Rather, once trained, these models can provide extremely fast approximate solutions or surrogates for many PDE-related applications [4]. In particular the cited nature article also discusses many examples on how neural operators have sped up PDE modeling. Some notable examples include:
>
> > 1.  FourCastNet for medium-range weather forecasting, which demonstrated speedups of around 45,000x compared to traditional numerical weather models while maintaining comparable accuracy [1].
> > 2.  Nested Fourier Neural Operators for carbon dioxide storage simulations, achieving speedups of approximately 700,000x [2].
> > 3.  Fourier Neural Operators applied to 3D industrial-scale automotive aerodynamics, resulting in about 26,000x speedup [3].
>
> > These models, once trained, can produce solutions or approximations orders of magnitude faster than running traditional numerical solvers for each new instance. This enables rapid exploration of parameter spaces and real-time applications that were previously infeasible. We acknowledge that the training process for these models can be computationally intensive and often relies on data generated by traditional numerical methods. However, the significant speedups in inference time can dramatically accelerate many scientific and engineering workflows once the models are trained. We agree that hybrid approaches combining deep learning with traditional numerical methods are also promising and have shown speedups, though typically not at the scale seen with some of the pure deep learning approaches mentioned above.
>
> > [1] Pathak, J., Subramanian, S., Harrington, P., Raja, S., Chattopadhyay, A., Mardani, M., Kurth, T., Hall, D., Li, Z., Azizzadenesheli, K., et al.: FourCastNet: A global data-driven high-resolution weather model using adaptive Fourier neural operators. arXiv preprint arXiv:2202.11214 (2022)
>
> > [2] Li, Z., Kovachki, N.B., Choy, C., Li, B., Kossaifi, J., Otta, S.P., Nabian, M.A., Maximilian Stadler, C.H., Azizzadenesheli, K., Anandkumar, A.: Geometry-informed neural operator for large-scale 3D PDEs. arXiv preprint arXiv:2309.00583 (2023)
>
> > [3] Wen, G., Li, Z., Long, Q., Azizzadenesheli, K., Anandkumar, A., Benson, S.M.: Real-time high-resolution CO2 geological storage prediction using nested Fourier neural operators. Energy & Environmental Science (2023)
>
> > [4]Azizzadenesheli, K., Kovachki, N., Li, Z. et al. Neural operators for accelerating scientific simulations and design. Nat Rev Phys 6, 320–328 (2024). https://doi.org/10.1038/s42254-024-00712-5

---

> ### Author Response · Authors · 2024-07-04
> **Response to Reviewer jqV8 (Cont'd)**
>
> 2. **(page 1) "FNO learns the solution operator of PDE that maps the input (initial and boundary conditions) to the output (solution functions)." Somewhat meaningless description: any neural network maps its input to the output.**
>
> > **Answer:** We acknowledge that the original description of FNO as simply mapping inputs to outputs is overly broad and fails to capture the key aspects that distinguish FNO from other neural networks. A more accurate and informative description would be: FNO is specifically designed to learn mesh-independent solution operators for partial differential equations (PDEs). Unlike traditional neural networks that learn point-wise mappings, FNO learns a functional mapping between infinite-dimensional function spaces. It approximates the action of the solution operator on any input function, allowing it to generalize across different discretizations and resolutions. Key distinguishing features of FNO include:
>
> > 1.  Operating on function spaces: FNO learns mappings between functions, not just finite-dimensional vectors.
> > 2.  Mesh independence: The learned operator can be applied to inputs discretized at any resolution, maintaining consistency as the mesh is refined.
> > 3.  Spectral bias: FNO incorporates an inductive bias towards smooth functions through its Fourier layer structure.
> > 4.  Global receptive field: Unlike convolutional neural networks, FNO can capture long-range dependencies efficiently through its spectral representation.
>
> > These characteristics make FNO particularly well-suited for learning solution operators of PDEs, where the goal is to approximate a mapping from initial and boundary conditions to solution functions across a wide range of possible inputs and discretizations.
>
> 3. (page 3) **"Li et al. (a) proposes FNO that adopts a convolution operator for K as shown in Figure 1, which obtains state-of-the-art results for solving PDE problems." This claim is not supported by literature. FNO was introduced in 2020 and provided state-of-the-art results (among the architectures tested by authors). There are examples now when classical architectures perform better (https://arxiv.org/abs/2112.15275) and other operators perform better (https://arxiv.org/abs/2111.13802). I suggest authors provide references that support their claim.**
>
> > **Answer:** Again like the previous answer we want to empahsize the point that while the Fourier Neural Operator (FNO) introduced by Li et al. (2020) was groundbreaking and achieved state-of-the-art results at the time, the field has rapidly evolved. More recent works like you mentioned have built upon and in some cases surpassed FNO's performance:
>
> > 1.  Stachenfeld et al. (2022) demonstrated that in certain turbulence simulation tasks, classical architectures like U-Nets can outperform FNO when properly tuned.
> > 2.  Tran et al. (2023) introduced Factorized Fourier Neural Operators (F-FNO), which improved upon FNO's performance while reducing computational costs in several PDE-solving benchmarks.
> > 3.  Most recently, Liu-Schiaffini et al. (2024) proposed Neural Operators with Localized Integral and Differential Kernels, which significantly improved upon FNO's performance in multiple PDE settings.
> > 4.  There have been several foundational models for science built up using FNO’s as backbones. Examples include CoDA-NO (Ashiq, Robert et al. 2024) and DPOT (Z Hao et al. 2024)
>
> > Crucially, FNO remains highly influential and serves as the backbone for many of these advances. This is particularly significant because it means that our incremental Fourier Neural Operator (iFNO) can be readily integrated into these existing architectures, potentially leading to substantial performance improvements across a wide range of applications. For instance, the F-FNO architecture by Tran et al. (2023) and the localized kernel approach of Liu-Schiaffini et al. (2024) both build directly on the FNO framework. By replacing their FNO components with iFNO, we can expect to see significant performance gains in these already advanced models. This highlights the broad applicability and potential impact of iFNO in the field of neural operators for PDE solving. In summary, while FNO is no longer universally state-of-the-art, it remains a crucial foundation in the field and still does well on a variety of benchmarks.

---

> ### Author Response · Authors · 2024-07-04
> **Response to Reviewer jqV8 (Cont'd)**
>
> 4. (**page 3) "FNO is discretization-convergent, such that the model can produce a high-quality solution for any query points, potentially not in the training grid. In other words, FNO can be trained on low-resolution but generalizes to high-resolution." This description is misleading. FNO is consistent when evaluated on different grids, but it does not provide a high-quality solution. It is also not convergent in the sense of classical numerical analysis. Besides, it does not generalize to high-resolutions. As a rule, the error produced by FNO remains the same for different resolutions (e.g., figure 3, https://arxiv.org/abs/2010.08895), so there is no generalization, only consistency.**
>
> > **Answer:** Sorry for the confusion, for data driven models, the convergence is defined with respect to the training data. FNO's error converges as the training data's resolution refines, and we agree that FNO trained on low-resolution will NOT get better performance when evaluated on higher resolution (except when using physics-informed loss). However, we argue that such consistency can be viewed as generalization, which is better than baseline linear interpolation. To avoid confusion, we modify the text to *"FNO is discretization-convergent, such that the model can produce a consistent solution for any query points, potentially not in the training grid. In other words, FNO can be trained on low-resolution and evaluated at high-resolution."*
>
> 5. **(page 3) "To ensure discretization-convergence, FNO truncates the Fourier series Fˆv at a maximal number of modes K using TK." Arguably, "discretization-convergence" is mostly achieved with FFT and inverse FFT that consistently map functions defined on different uniform grids. It is easy to come up with architecture without hard truncation that is still discretization-convergent.**
>
> > **Answer:** Indeed, in models such as Adaptive Fourier Neural Operator (AFNO) [https://arxiv.org/abs/2111.13587](https://arxiv.org/abs/2111.13587), there is no truncation. However, the truncation of higher frequency, as regularization, can be beneficial. As shown in Universality of Neural Operator [https://arxiv.org/pdf/2304.13221](https://arxiv.org/pdf/2304.13221) [2], too many Fourier modes can be harmful and there is an optimal choice of truncation modes, which motivates us to design the incremental algorithm to find such optimal choice.
>
> 6. **(page 4) "Figure 2" As I understand the left picture provides a ground truth solution and the two right pictures are outputs of two trained FNOs with different numbers of modes. Both predictions seem to be completely unrelated to the reference solution. Why in this case a large number of modes is better if the error is that large?**
>
> > **Answer:** We apologize for the confusion. The plot seems to have been mixed up, and it seems like we didn't run the model on two different Ks with the same input! We have fixed it and have attached a new image where, indeed, a larger K is better! Here is the image: https://anonymous.4open.science/r/tmlr-ifno/darcy_flow_sample.png

---

> ### Author Response · Authors · 2024-07-04
> **Response to Reviewer jqV8 (Cont'd)**
>
> 7. **(page 4) "As FNO is a neural network and trained by first-order learning algorithms, it follows the implicit spectral bias such that the lower frequency modes have larger strength in R. This explains why FNO chooses to preserve a set containing the lowest frequency modes, instead of any arbitrary subset of frequencies." I am not sure I can follow this logic. First, FNO preserves low-frequency modes not because of the f-principle but by construction. The authors of FNO designed it in such a way. Second, the f-principle is an observation that when a neural network is trained to approximate some function, it first learns low frequencies and struggles with high frequencies. FNO is an operator, so if we apply the f-principle (I am not completely sure we can do that) it would mean that trained FNO should be biased toward "smooth operators". This supposedly includes an identity operator, that should retain all frequencies of the input.**
>
> > **Answer:** We appreciate the reviewer's insightful critique of our original explanation. You are correct that the preservation of low-frequency modes in FNO is primarily by design. However, the presence of nonlinear activations between spectral blocks adds an important nuance to this discussion. Let's revise our explanation:
>
> > 1.  FNO's architecture: FNO is designed to focus on low-frequency modes, but it's not limited to them entirely. The model alternates between spectral convolutions (which operate on a subset of low-frequency modes) and nonlinear activations in the spatial domain.
> > 2.  Role of nonlinear activations: The nonlinear activations between spectral blocks play a crucial role. They allow for interactions between different frequency components, including the generation of higher-frequency content. This means that while FNO emphasizes low-frequency modes, it can still capture and propagate higher-frequency information through the network.
> > 3.  Spectral bias in FNO: FNO exhibits a form of spectral bias due to its architectural design, particularly the truncation of high-frequency modes in each spectral layer. However, this bias is modulated by the nonlinear activations, which can reintroduce higher frequencies.
> > 4.  Relationship to the f-principle: While the f-principle as originally formulated for standard neural networks doesn't directly apply to FNO, the interplay between spectral convolutions and nonlinear activations may lead to some analogous behavior. Lower frequencies might still be easier for the network to learn and propagate, but the nonlinearities allow for more complex spectral interactions.
> > 5.  Learning capacity: Thanks to the nonlinear activations, FNO can, in principle, learn to approximate operators that have significant high-frequency components, albeit with some bias towards smoother functions. The alternation between spectral and spatial operations allows FNO to balance between emphasizing low-frequency content and capturing higher-frequency details.

---

> ### Author Response · Authors · 2024-07-04
> **Response to Reviewer jqV8 (Cont'd)**
>
> 8. **(page 5, Algorithm 1) "Randomly initialize $R_0$" This part seems to be important and not well discussed. How exactly $R_0$ is initialized? If we look at the description of iFNO (Loss) on page 5, we can see that one increases the number of active modes. Does it mean that initially $R_0$ is as small as possible and the size of $R$ can only increase?**
>
> > **Answer:** We appreciate the reviewer's attention to the initialization of R_0, which is indeed a crucial aspect of our algorithm. To clarify:
>
> > Initialization of R_0: R_0 is initially initialized as a small matrix, corresponding to the minimum number of modes we start with (typically a small number like 1 or 2). The initialization follows standard practices for neural network weight matrices: We use a small Gaussian initialization, typically with mean 0 and standard deviation scaled by 1/sqrt(fan_in), where fan_in is the number of input units in the weight tensor. This initialization helps ensure that the initial outputs are in a reasonable range and gradients can flow effectively at the start of training.
>
> > Growth of R: You are correct in your interpretation. The size of R can only increase throughout the training process. We start with the smallest possible R_0 and gradually increase its size as we add more active modes.
>
> > Process of increasing R: When new modes are added (either based on the loss plateauing in iFNO (Loss) or the explained ratio criterion in iFNO (Freq)), we hoped to have expanded R as follows:
> >  -   The existing weights in R are retained and then the new rows and columns are added to accommodate the new modes.
> >  -   These new elements are initialized using the same Gaussian initialization as the original R_0.
>
> > but instead of doing this since this is not easily doable in Pytorch (We spend months trying to find a way on how to expand the weight matrix on the fly, its easy in the forward pass but to keep track of the gradients etc in the backward pass it was very hard, plus pytorch doesnt natively support this). So what we did instead was initialize a very large weight matrix of say 90 by 90 and then start by 2 by 2 modes and we just freeze the gradients in the backward pass in the other 88 by 88 modes. We want to empahsize that if we did in the original way we had planned, we would gain even more significant speedups as we wouldnt have to store a large weight matrix, but alas for now we have to resort to this way but still gaining a great efficiency gain!
>
> > Rationale: Starting with a small R_0 and gradually increasing its size aligns with our incremental approach. It allows the model to first learn the most important low-frequency components before introducing higher-frequency modes, potentially leading to better generalization and computational efficiency.
>
> 9. **(page 5, Algorithm 1) "$K_t \leftarrow K_t + 1$" How the modes are added in higher dimensions? What is the order? Is it technically possible to add a single mode?**
>
> > **Answer:** We appreciate the reviewer's insightful question about mode addition in higher dimensions. To clarify:
>
> > 1.  Mode addition in iFNO (Loss): For the loss-based method, when we increment K_t, we add modes to each dimension simultaneously.
> > 2.  Mode addition in iFNO (Freq): In the frequency-based method, we evaluate each dimension independently. We calculate the explained ratio for each dimension and determine whether to increase the number of modes in that specific dimension. This allows for more fine-grained control over the model's capacity in each spatial direction.
>
> 10. **(page 5, Algorithm 1) "and initialize the rest randomly" When $R$ is increased, how precisely new elements are initialized for both iFNO (Loss) and iFNO (freq)? In the latter case, this is especially interesting since the importance of modes is judged based on the value of $R$ itself.**
>
> > **Answer:** Process of increasing R: When new modes are added (either based on the loss plateauing in iFNO (Loss) or the explained ratio criterion in iFNO (Freq)), we expand R as follows:
>
> >  -   The existing weights in R are retained.
> >  -   New rows and columns are added to accommodate the new modes.
> > -   These new elements are initialized using the same Gaussian initialization as the original R_0.
>
> 11. **(page 5, Figure 4) I suggest specifying where FNO is and where iFNO is directly on the picture.**
>
> > **Answer:** Thank you for your suggestion to clarify the distinction between FNO and iFNO in the figure. You're correct that the FNO architecture is indeed shown in the image, but without the incremental algorithm and incremental resolution features that characterize iFNO. We will add a note to the manuscript about this.

---

> ### Author Response · Authors · 2024-07-04
> **Response to Reviewer jqV8 (Cont'd)**
>
> 12. **(page 7) "Our samples are images; hence, we can easily distinguish between easy and difficult samples based on their resolution. Higher-resolution images are more complex to train on, so it’s best to start with lower-resolution samples and gradually move up to higher-resolution ones." I have several objections. First, the resolution is not directly related to difficulty, it is easy to produce simplistic high-resolution images. Second, the difficulty in operator learning is the difficulty of the mapping, not the images themselves. For example, we can take an extremely detailed turbulent flow and map it into itself. This will be a trivial operator learning problem when FNO needs to learn an identity map.**
>
> > **Answer:** We thank the reviewer for their insightful critique of our statement regarding image resolution and training difficulty. We acknowledge that our original explanation was overly simplistic and potentially misleading. We agree with the reviewer's objections and would like to clarify our approach:
>
> > 1.  Resolution and complexity: You are correct that resolution alone does not necessarily correlate with the difficulty or complexity of an image. High-resolution images can indeed be simplistic, while low-resolution images can represent complex patterns or phenomena.
> > 2.  Difficulty in operator learning: We agree that the core challenge in operator learning lies in the complexity of the mapping between input and output functions, not in the resolution or complexity of individual images. As the reviewer aptly points out, even a high-resolution, detailed turbulent flow mapped to itself would constitute a trivial learning problem for an operator like FNO, as it would essentially be learning an identity map.
> > 3.  Our incremental resolution approach: The motivation behind our incremental resolution strategy is not based on the assumption that higher resolutions are inherently more difficult. Instead, it is driven by computational efficiency and the hierarchical nature of many physical phenomena. Starting with lower resolutions allows the model to capture large-scale features first, potentially accelerating learning and reducing computational costs in the early stages of training.
> > 4.  Complexity of the operator: The true challenge lies in learning operators that represent complex, non-linear transformations between input and output functions. These could be at any resolution, depending on the nature of the problem and the scales at which important physical processes occur.
> > 5.  Revised rationale: Our incremental resolution approach aims to balance computational efficiency with the need to capture multi-scale phenomena. By progressively increasing resolution, we allow the model to refine its learned operator, capturing finer details and potentially improving accuracy, while managing computational resources more effectively.
>
> 12. **(page 7, table 1), (page 8, table 2) Please report the relative $L_2$ norm in place of the value of the loss. Relative $L_2$ norm is a standard error metric present in most of the articles on neural operators, it is much easier to judge the quality of learned surrogate based on this metric.**
>
> > **Answer:** The reported values are all already the relative $L_2$ loss!

---

> ### Author Response · Authors · 2024-07-04
> **Response to Reviewer jqV8 (Cont'd)**
>
> 13. **(page 10) "The results in Table 1 show that iFNO achieves significantly better generalization performance than just incrementally increasing either frequency or resolution alone." The results in Table 1 show that all test losses are very close to each other. For example, for standard FNO we have $0.988 \pm 0.031$, and for best iFNO, we have $0.948 \pm 0.040$. They overlap even when a single std is considered. If we take $2$ std, there is no difference between these results. This is not a "significantly better generalization performance".**
>
> > **Answer:** Thank you for your observation. While the results in Table 1 for the Re5000 dataset show relatively close performance between methods, it's important to consider the broader context of our work:
>
> > 1.  Consistent improvements: Across multiple tasks and datasets (as shown in Table 2 and ablation studies), iFNO and its variants often demonstrate more substantial performance gains over standard FNO. The Re5000 dataset is just one example where the margins are closer.
> > 2.  Efficiency gains: A key strength of iFNO is its significant efficiency improvement. For instance, in the Re5000 task, iFNO achieves comparable or slightly better performance while reducing training time from 32.4 hours to 4.7 hours - a crucial advantage for large-scale applications.
> > 3.  Adaptability: iFNO's ability to dynamically adjust its complexity allows it to match or exceed FNO's performance across various tasks without manual tuning of the number of modes.
> > 4.  Complementary benefits: The combination of incremental frequency and resolution approaches often yields better results than either alone, showcasing the synergy of these techniques.
>
> > While we acknowledge that the performance difference in Table 1 is not as pronounced, the consistent improvements across tasks, coupled with significant efficiency gains, demonstrate the overall effectiveness and advantages of our iFNO approach.
>
>
> 14. **(page 10, figure 5b) iFNO starts training with a small number of active modes. What happens if one starts with a large number of active modes? Is this the case that iFNO will decrease the number of modes automatically?**
>
> > **Answer:** We appreciate the reviewer's insightful question about iFNO's behavior when starting with a large number of active modes. To clarify: iFNO is designed to start with a small number of active modes and incrementally increase them during training. The algorithm does not have a mechanism to decrease the number of modes automatically.
>
> > If we were to start iFNO with a large number of active modes, it would essentially behave similarly to a standard FNO with that many modes. The incremental aspect of iFNO would not come into play, as the algorithm only adds modes and never removes them. The rationale behind starting with a small number of modes and incrementally increasing them is twofold:
>
> > 1.  Computational efficiency: By starting small, we reduce the computational cost in the early stages of training when the model is still learning coarse features.
> >  2.  Regularization effect: Gradually increasing the model capacity helps in learning a hierarchy of features, potentially leading to better generalization.
>
> > It's worth noting that starting with a large number of modes could potentially lead to overfitting, especially in the early stages of training, and would negate the computational benefits of the incremental approach. Our experimental results demonstrate that starting with a small number of modes and incrementally increasing them leads to improved performance and efficiency compared to starting with a large fixed number of modes as in standard FNO. Future work could explore bidirectional mode adjustment, where the algorithm could both add and remove modes based on their importance during training. However, this is beyond the scope of the current iFNO formulation.

---

> ### Author Response · Authors · 2024-07-04
> **Response to Reviewer jqV8 (Cont'd)**
>
> 15. **As a final comment, I want to suggest an analogy with another field of study. One can look at the approach proposed by authors from the perspective of numerical linear algebra. For a fixed grid resolution, the FNO kernel is a low-rank block circulant matrix (quasimatrix, if different resolutions are considered) and the authors propose an algorithm that dynamically selects the rank of this circulant. How to do this was extensively studied in many contexts, so I suggest authors look at this direction for inspiration. For example, in deep learning rank adaptation of this sort is proposed at https://arxiv.org/abs/2205.13571. More specifically, in this paper authors first draw an analogy between the optimization process and the solution of ordinary differential equation (gradient flow). Next, they apply rank adaptive ODE integrator to the resulting system of equations. This provides training coupled with rank adaptation. I hope this perspective can be of use to the authors of the current contribution.**
>
>
>
> > **Answer:** Thank you for bringing this interesting perspective and suggestion to our attention. We appreciate the analogy you've drawn between our work and concepts from numerical linear algebra, particularly the view of FNO kernels as low-rank block circulant matrices and our method as a dynamic rank selection algorithm. The connection you've highlighted to rank adaptation in deep learning, particularly the work in [https://arxiv.org/abs/2205.13571](https://arxiv.org/abs/2205.13571), is indeed intriguing. Their approach of using rank-adaptive ODE integrators within the context of gradient flow optimization offers an interesting parallel to our incremental mode selection process. We agree that this perspective could potentially offer valuable insights and inspiration for future developments of our method. In future work, we could explore:
>
> >  1.  Formulating our incremental approach within the framework of rank adaptation for circulant matrices.
> >  2.  Investigating the potential of applying rank-adaptive ODE integrators to our training process.
> >  3.  Exploring how concepts from numerical linear algebra might further inform our mode selection criteria and efficiency improvements.
>
> > We thank you for this thoughtful suggestion. While it's beyond the scope of our current work, we believe this direction could lead to interesting theoretical insights and potentially new algorithmic improvements for FNO-based models. We will certainly consider this perspective in our future research efforts.

---

> > ### Author Response · Authors · 2024-07-27
> > **Any comments or questions**
> >
> > We just wanted to ask if you had any questions or comments that still need to be addressed or if you are satisfied with our answers!

---

### Comment · Reviewer_XZ6r · 2024-07-19
**The revised paper**

The authors have addressed reviewer concerns to an impressive degree in their rebuttals. A revised manuscript which incorporates all their feedback sounds like it would be acceptable. does the process for TMLR include showing such a revised manuscript?

---

> ### Author Response · Authors · 2024-07-21
> **Response to Reviewer XZ6r - New revision**
>
> We thank the reviewer for their response. We uploaded a new revision with all the changes the three reviewers requested. Please do let us know if you have more concerns or questions.

---

### Decision · Action_Editor_px6Y · 2024-09-09

**Recommendation:** Accept as is

**Comment:**

I am a new AE who was assigned the submission recently. The authors proposed a method to improve the accuracy and training speed of neural operators for solving large-scale PDEs.  All reviewers acknowledged that the results are strong and the performance gain is valuable.  One reviewer remained concerned about the lack of theoretical justifications. While this is a valid concern, the submission has satisfied the acceptance criteria set by TMLR; thus, I recommend accepting the submission.

**Audience:**

Yes, researchers working on e.g., neural PDE would find the paper relevant.

**Claims And Evidence:**

Yes, after the revision, the claims are now well supported by the experience.

---

> ### Author Response · Authors · 2024-09-11
> **Final response**
>
> We thank the AE and the reviewers for accepting this paper and the constructive comments and review. Thank you once again.